# Formation and geophysical character of transitional crust at the passive continental margin around Walvis Ridge, Namibia

Gesa Franz[1], Marion Jegen[1], Max Moorkamp[2], Christian Berndt[1], Wolfgang Rabbel[3]

[1]GEOMAR Helmholtz Centre for Ocean Research Kiel, Germany
[2]Ludwig-Maximilians University of Munich, Germany
[3]Christian-Albrechts University Kiel, Germany

*Correspondence to*: Gesa Franz (gfranz@geomar.de)

**Abstract**

When interpreting geophysical models, we need to establish a link between the models' physical parameters and geological
units. To define these connections, it is crucial to consider and compare geophysical models with multiple, independent parameters. Particularly in complex geological scenarios, such as the rifted passive margin offshore Namibia, multi-parameter analysis and joint inversion are key techniques for comprehensive geological inferences. The models resulting from joint inversion enable the definition of specific parameter combinations, which can then be ascribed to geological units. Here we perform a user-unbiased clustering analysis of the two parameters electrical resistivity and density from two models
derived in a joint inversion along the Namibian passive margin. We link the resulting parameter combinations to break-up related lithology, and infer the history of margin formation. This analysis enables us to clearly differentiate two types of sediment cover. The first type of sediment cover occurs near-shore and consists of thick, clastic sediments, while the second type of sediment cover occurs further offshore and consists of more biogenic, marine sediments. Furthermore, we clearly identify areas of interlayered massive, and weathered volcanic flows, which are usually only identified in reflection seismic
studies as seaward dipping reflectors. Lastly, we find a distinct difference in the signature of the transitional crust south of- and along the supposed hot-spot track Walvis Ridge. We ascribe this contrast to an increase in magmatic activity above the volcanic centre along Walvis Ridge, and potentially a change in the melt sources or depth of melting. This change of the predominant volcanic signature characterizes a rift-related southern complex, and a plume-driven Walvis Ridge regime. All of these observations demonstrate the importance of multi-parameter geophysical analysis for large-scale geological
interpretations. Additionally, our results may improve future joint inversions using direct parameter coupling, by providing a guideline for the complex passive margins parameter correlations.

## 1 Introduction

Passive continental margins can be grouped into the end member types "magma-poor" or "non-volcanic", and "magma-dominated" or "volcanic" rifted margins (Blaich et al., 2011; Pérez-Gussinyé & Reston, 2001; Peron-Pinvidic et al., 2013).
The two types of margins feature distinct crustal characteristics, which include extensively rifted continental crust and

serpentinized mantle for the magma-poor-, and surface volcanic flows, as well as crustal intrusions for the magma-dominated margin (Blaich et al., 2011; Sawyer et al., 2007). Intermediary margin types comprise both end member types' features of different extents, for example a wide extension zone with a subsequent narrow magmatic zone (Clift and Lin, 2001a; Clift and Lin, 2001b; Franke, 2013; Zhu et al., 2012). In geophysical investigations, passive margins are often classified based on seismic surveys and seismostratigraphy (e.g. Elliott et al., 2009; Franke, 2013; Jackson et al., 2000; Planke & Eldholm, 1994; Planke et al., 2000). In these surveys, magma-poor passive margins are characterized by shallow reflectors, normal faults and deep syn-rift basins which image the hyperextended, thinned continental crust (Blaich et al., 2011; Franke et al., 2013), as well as high upper mantle velocities imaging mantle serpentinization with no drastic Moho transition (Brune et al., 2017; Minshull, 2009; Reston & Pérez-Gussinyé, 2007; Whitmarsh et al., 2001). Magma-dominated margins are identified by thickened crust and high velocity underplating imaging the intra-crustal magmatic input, as well as seaward dipping reflectors imaging surface volcanic flows (Eldholm et al., 2000; Franke, 2013; Planke et al., 2000). Only limited electromagntic surveys have been performed to image the electrical resistivity structure of passive margins. Authors have identified varying structures, such as a conductive lithospheric anomaly imaging mantle upwellings (Attias et al., 2017); upper crustal conductors imaging sub-basalt sediment basins, or (Hoversten et al., 2015) or graphite and massive sulphide mineralizations (Corseri et al., 2017; Heinson et al., 2005); and high lower crustal resistivities depicting magmatic underplating (Jegen et al., 2016). The underlying reason for the different characteristics is linked to the "Plates vs Plumes" hypotheses, which discusses the triggers of continental break-up in terms of mantle convection, mantle plumes, melt generation, and plate tectonics (Anderson, 2001; Foulger, 2010; Morgan, 1971; Wilson, 1963). The two end-member models ascribe volcanic anomalies and continental rifting to either pure plate tectonic stresses, and upper mantle convection ("plate theory"), or full mantle convection, and deep mantle plumes originating from the core-mantle boundary ("plume theory").

The Walvis Ridge and the Namibian margin have long been a focus of research in terms of passive margin formation and mantle plume-lithosphere interaction (e.g. Dingle & Simpson, 1976; Eldholm et al., 2000; Fromm et al., 2015; Ryberg et al., 2015; Yuan et al., 2017). Many regional geophysical studies around Walvis Ridge have investigated typical breakup related geological features such as flood basalts, thickened crust, magmatic underplating, and various crustal intrusions (e.g. Bauer et al., 2003; Fromm et al., 2017a; Gladczenko et al., 1998; Goslin & Sibuet, 1975; Kapinos et al., 2016; Planert et al., 2017; Ryberg et al., 2022), which are reviewed in detail in the next section. While Walvis Ridge is often referred to as a classic plume example (White & McKenzie, 1989), different authors have also argued, that the amounts of magmatism and thermal alteration of the crust at the Namibian coast are too small to justify a continental breakup induced by the arrival of a large plume head (Fromm et al., 2017a; Koopmann et al., 2016; Ryberg et al., 2022).

In this study, we complement previous geophysical studies by analysing combined parameter models of electrical resistivity and density. The models, which form the basis for this study, are published in Franz et al. (2021), where data, model resolution, and inversion scheme are presented in detail. Here, we focus on a clustering analysis of the relationship of density

and electrical resistivity from two models derived in a cross-gradient coupled joint inversion. This joint inversion approach addresses the ambiguity arising from single method inversion, meaning that no unique parameter-lithology correlations exist for electrical resistivity and density. The clustering concept augments and validates previous model interpretations by providing an independent, user-unbiased spatial parameter correlation. The mapping of these two parameters and comparison of the defined clusters with independent geophysical models and logging data from marine boreholes, as well as

reconstructing the temporal margin evolution, enables us to link the clusters to geological units and margin formation stages. This approach complements existing research concerning the breakup mechanisms of the South Atlantic, because the integration of the two parameters electrical resistivity and density improves definition of geological units, and thus advances the interpretation of typical breakup related lithology. Additionally, the characterization of clusters in the electrical resistivity – density correlation, can be a guideline for future improvement of joint inversion using direct parameter coupling

(Moorkamp et al., 2017).

## 2 Geological Setting

### *Phases of the geodynamic evolution of the Namibian passive continental margin*

The late Triassic breakup of the mega-continent Pangea comprises the separation between Gondwana and Laurentia and the rupture of Gondwana in a western (Africa and South America) and eastern (Australia, Antarctica, Madagascar, and India)

block (Moulin et al., 2010; Seton et al., 2012).

Continental breakup of western Gondwana started in the south with early rifting in the late Jurassic (~140-145 Ma) and propagated northwards until it reached the area around present-day northern Namibia at around 127-133 Ma (Heine et al., 2013; Macdonald et al., 2003; Nürnberg & Müller, 1991; Torsvik et al., 2009). Rupture was likely halted at structures such

as the Falkland-Agulhas- and Florianopolis fracture zones leading to a segmented rift propagation (Franke et al., 2007; Jackson et al., 2000; Koopmann et al., 2014). The northward propagation of the continental breakup strongly correlates with different phases of the margin evolution, which is outlined in the following sections.

### 2.1 Phase 1: Early continental rifting

The crust created by the initial northward propagation of continental rifting is characterized by its original background rock, an abundance of different sized fractures and faults, and aligned rift valleys or (half-)graben structures (Clemson et al., 1997; Glen et al., 1997). The West-African basement rock is built up from the Precambrian Congo Craton, as well as the Pan-African Kaoko- and Damara fold belts (Begg et al., 2009; Frimmel, 2009; Haas et al., 2021, see Fig. 1). Re-activated faults in these fold belts also pose the main extensional centres of the Jurassic rift stage. They trend north-south in the Kaoko belt

and southwest-northeast in the Damara belt (Clemson et al., 1997; Passchier et al., 2002; Salomon et al., 2017). This alignment of Cretaceous rift valleys with the Pan-African fold belts emphasizes the importance of pre-rift basement tectonics on the Cretaceous rift geometry (Gassmöller et al., 2016).

The amount of influence of the Tristan mantle plume in the initiation of continental rifting is still a matter of debate (Gibson
et al., 2006; Turner et al., 1996; White & McKenzie, 1995). However, there is evidence that rifting took part prior to magmatic activity (Glen et al., 1997), and that the initial magmatic activity are mainly of upper mantle composition instead of a deep mantle plume source (Franke, 2013; Peate, 2013). Both point to the possibility that the Tristan plume matured as a consequence of rifting and emanated due to the weakened rifted crust (Franke, 2013). Burke & Dewey (1973) initially described the interaction of mantle plumes and triple junctions. Their theories pose a possible explanation for the maturation
of the Tristan plume at its proposed location, through formation of a rifted triple junction along the weak zones of the re-activated faults of the Kaoko-, Damara-, and Gariep fold belts (Franke, 2013; Trumbull et al., 2004). Alternative models explain the entire continental magmatic series without lithospheric mantle input but only crustal contamination (Ewart et al., 2004; Thompson et al., 2001). This indicates an early impact of the rising Tristan plume and therefore a strong influence of the plume in rift generation (Buiter & Torsvik, 2014; Courtillot et al., 2003; Foulger, 2010).

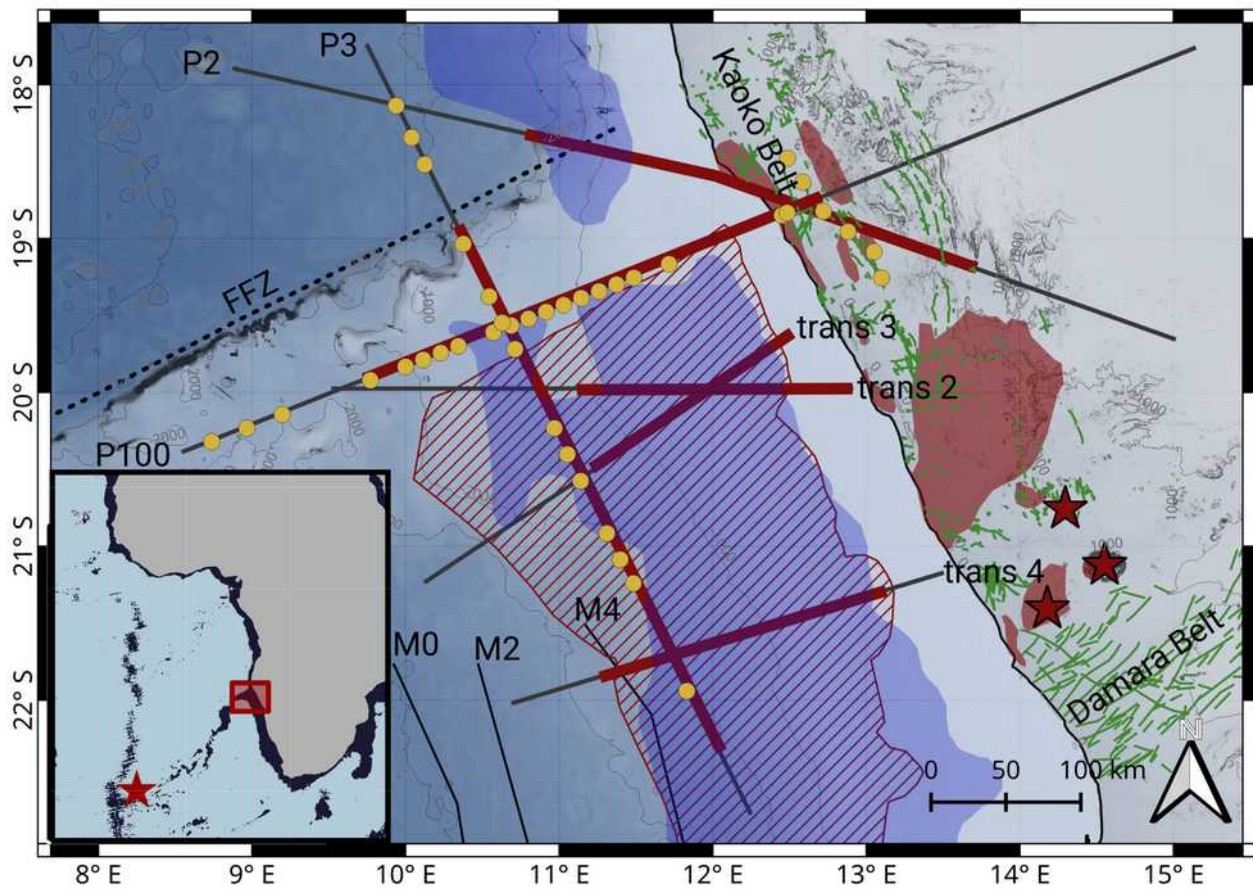

**Figure 1: Topographic map of the survey area with geophysical stations and profiles, as well as geological-, and tectonic features. Yellow circles are marine MT stations, grey lines are seismic profiles, where red marks areas of interpreted high velocity magmatic underplating. Studies are performed by Fromm et al. (2017a, 2015) (profiles P100 and P2), Planert et al. (2017) (profile P3), and Gladczenko et al. (1998). (profiles trans 2 – trans 4). Red dashed area is the outline of seismically imaged seaward dipping reflections (SDR) (Koopmann et al., 2016), black dashed line is the Florianopolis fracture zone (FFZ). Blue area marks the outline of sediment cover thicker than 3 km, taken from Maystrenko et al. (2013). Red areas onshore are the Etendeka continental flood basalts, red stars mark the Brandberg, Messum, and Doros intrusive complexes (Owen-Smith et al., 2017; Teklay et al., 2020). Green lines represent magmatic dykes of the Kaoko and Damara belts (Salomon et al., 2017; Trumbull et al., 2004; respectively). Magnetic lineations M0 to M4 are taken from Moulin et al. (2010). In the map inlay, the read box marks the large map's extent and the red star is the location of the Tristan hotspot. Dark areas are bathymetric features, where water depth is less than 3000 m, the entire hot spot track of Walvis Ridge clearly links the Namibian landfall to the mid-ocean ridge.**

## 2.2 Phase 2: Arrival of magmatism

In the continental areas of Namibia, an abundance of intrusive and extrusive structures reflect the pre-breakup arrival of magmatism. Fracturing and thinning of the rifting crust eventually leads to decompression melting and rise of magma (Begg

et al., 2009; Foucher et al., 1982; Franke et al., 2007; McKenzie & Bickle, 1988), which forms three different kinds of features: a) intrusive magmatic bodies, b) magmatic dykes and sills, and c) continental flood basalts (CFB).

On the Namibian margin, examples for the first kind of feature (a) are the Jungfrau- and Sargdeckel-, Messum-, Brandberg-, and Doros complexes located in the southernmost part of the exposed Etendeka magmatic province (Teklay et al., 2020; Harris et al., 1999; Schmitt et al., 2000; Owen-Smith et al., 2017; respectively, see Fig. 1). All of them are intrusive, igneous rocks of mixed composition, including depleted, shallow mantle as well as enriched, deeper, plume-like components, indicating an interaction of the Tristan mantle plume with the overlying lithosphere (Ewart et al., 2004; Thompson et al., 2001). Despite their varying chemical composition, they are all massive magmatic bodies, which were trapped in the crust and have now emerged to the surface.

Melts, that were not trapped in the described massive intrusive bodies, may rise to the surface as magmatic dykes (b). Examples on the Namibian margin are the Huab-, and Henties Bay-Outjo dyke swarms, as well as dykes oriented along the Kaoko Belt (Duncan et al., 1989; Keiding et al., 2013; Trumbull et al., 2004; Salomon et al., 2017; respectively, see Fig. 1). The orientation of these dykes correlates with the faults and fractures described in Sect. 2.1. Therefore, melts ascended in the pathways created by preceding continental rifting and may be accompanied by horizontally emplaced sills.

These magmatic dykes then form the feeding system for the massively extruded continental flood basalts (c) which are asymmetrically distributed over the African- and South American margin as the Etendeka- and Paraná CFB provinces, respectively (Peate, 2013). Possible reasons for the asymmetry include the position of the Tristan mantle plume below the Paraná province (O'Connor & Duncan, 1990), an uneven lithosphere base (Gassmöller et al., 2016; Thompson & Gibson, 1991), the pronounced topography on the African side through uplift hindering eastward magmatic flow (White & McKenzie, 1989), or a sheared extension leading to a rotation of South America (Aslanian et al., 2009; Peate, 2013)

### 2.3 Phase 3: Transition from rifting to continental breakup

A clear definition of the transition from rifting to breakup (continent-ocean-transition, COT) is generally difficult. The transition is a continuous process, leading to a zone of highly variable transitional crust. This crust usually consists of the initially rifted and intruded continental crust described above, followed by an increase of dykes and horizontal sill intrusions, and eventually magmatic flows on the surface, which may appear consistently or periodically (e.g. Eldholm et al., 2000). A distinct continent-ocean-boundary (COB) is usually defined by the onset of submarine spreading, characterized by magnetic linear anomalies and seismic structure (Gladczenko et al., 1998; Rabinowitz & Labrecque, 1979). At the Namibian margin, the Cretaceous magnetic quiet zone coincides with continental breakup, impeding a clear COB mapping by magnetic lineations (Koopmann et al., 2016; Moulin et al., 2010; Seton et al., 2012). Additionally, different mechanisms for magnetic

lineation generation have been described, e.g. a linear pattern masked by overlying subaerial sheet flows, which may challenge this classical definition (Collier et al., 2017).

The increase of magmatic activity leads to an abundance of intrusive igneous material, referred to as magmatic underplating, which is usually associated with high velocity, high density lower crustal bodies (Becker et al., 2014; Fromm et al., 2017a; Gernigon et al., 2004; Hirsch et al., 2009; Mjelde et al., 2007, see Fig. 1). They may be produced by an accumulation of mantle melts below the crust due to a contrast in density, or by massive intrusions into the heated and weakened lower crust
(Eldholm et al., 2000; Franke, 2013; White et al., 1987).

Contemporaneously, the high  amount of melts leads to surface flows.  These surface flows at the transition from rifting to drifting represent themselves by sequences of seaward dipping reflectors (SDR) in marine seismic studies (Franke, 2013; Mutter et al., 1982; Paton et al., 2017; Planke et al., 2000, see Fig. 1), and as a wide, chaotic magnetic anomaly in magnetic
studies (anomaly G, or LMA for large magnetic anomaly in Rabinowitz & Labrecque, 1979; Moulin et al., 2010, respectively). This anomaly results from the SDR's periodic emplacement and may overlap with the classic COB marked by underlying magnetic lineations (Collier et al., 2017; Paton et al., 2017; Planke et al., 2000). SDR's are defined as series of compact volcanic flows interbedded with sediments, tephra, or hyaloclastites and may be emplaced subaerially or subaqueously in shallow marine environments (Jackson et al., 2000; Planke et al., 2000). The different depositional
environment  distinguishes them from the initial continental flood basalts (McDermott et al., 2018).

### 2.4 Phase 4: Halted breakup at the Florianopolis fracture zone

The crustal structure of Walvis Ridge changes strikingly rapidly at the Florianopolis Fracture Zone (FFZ, also referred to as Rio Grande fracture zone, see Fig. 1) (Goslin & Sibuet, 1975; Planert et al., 2017). While thickened crust and the features described above (magmatic underplating, periodic magmatic flows, and magmatic dykes) characterize the COT zone south
of Walvis Ridge, the crust north of the FFZ is distinctly thinner, with significantly less magmatic overprint compared to south of Walvis Ridge (Aslanian et al., 2009; Blaich et al., 2011; Planert et al., 2017).

Continental breakup around Walvis Ridge is dated at ~133 Ma, while breakup north of the FFZ took place at around 112 Ma, leading to ~20 Ma of halted spreading (Heine et al., 2013; Moulin et al., 2010). A likely reason for this pause is a change in
plate motion around this time, from a west-east, to a southwest-northeast extension (Heine et al., 2013; Koopmann et al., 2016; Moulin et al., 2010) as well as along margin variations in extension velocity (Brune et al., 2014). Most likely, the halted breakup plays an important role in the formation of an abundance of adjacent magmatic features like the thickened crust below Walvis Ridge and massive packages of interbedded flows (SDR) (Bauer & Schulze, 1996; Fromm et al., 2017a; Gladczenko et al., 1998). These pronounced structures caused by a prolonged plume-ridge interaction, as well as the

increased extension in the south may have prevented northward horizontal flow of the rising plume material (Georgen & Lin, 2001; Morgan et al., 2020; Planert et al., 2017).

**2.5 Phase 5: Ridge jump and continued continental breakup**

The maximum melt production around Walvis Ridge is dated at ~122 Ma, which marked the arrival of the plume tail at the
195 base of the lithosphere and consequently the decrease of magmatism (Gassmöller et al., 2016). Subsequently, an eastward ridge-jump moved the centre of breakup directly adjacent to the Angolan coast (~112 Ma), which lead to an abrupt transition between continental and oceanic crust (Aslanian et al., 2009; Moulin et al., 2010). The delayed breakup lead to considerably more continental rifting (width of ~350 km) north of the FFZ compared to the maximum of ~100 km south of the boundary (Torsvik et al., 2009). While the crust north of the FFZ is less affected by the plume material (Gassmöller et al., 2016),
Aptian salt basins further north are evidence for a preceding, syn-rift shallow marine environment (Blaich et al., 2011; Heine et al., 2013; Torsvik et al., 2009). For the preceding reasons, the continental margin north of the FFZ may be referred to as "magma-poor" (Blaich et al., 2011).

Following the continental break-up up to ~100 Ma, rapid cooling and flexural uplift were accompanied by increased erosion,
along the great escarpment (Braun, 2018; Gallagher & Brown, 1999; Margirier et al., 2019; Rust & Summerfield, 1990). In the late Cretaceous, the regional uplift of the South African or Kalahari Plateau- and the corresponding reactivation of the Precambrian shear zones lead to additional denudation along the margin (Baby et al., 2018; Brown et al., 2014; Raab et al., 2005; Rouby et al., 2009). Both of these mechanisms lead to significant (6-12 km) sediment deposits in the offshore Walvis, Luderitz, and Orange Basins. During the Palaeocene, increasing temperatures and humid climate likely increased the erosion
of the onshore break-up related Etendeka flood basalts, while the sediment supply to the offshore basins declined (Margirier et al., 2019, Braun et al., 2014; Baby et al., 2018). An explanation for this discrepancy may be the enhanced chemical erosion of the easily weathered basalts (Baby et al., 2018; Margirier et al., 2019). The most recent phase denudation, 35 – 0 Ma according to Margirier et al. (2019), and ~17 – 0 Ma according to Baby et al. (2018), is characterized by a major aridification and stable tectonic environment, and thus low sedimentation rates.

**3 Method**

In order to identify specific parameter ranges and -relationships representing the different stages of continental breakup at the Namibian margin, we analyse electrical resistivity and density models which have been derived from a joint data inversion with cross-gradient coupling, based on marine magnetotelluric- (MT) and satellite gravity data, respectively (Franz
et al., 2021). The joint inversion algorithm by Moorkamp et al. (2011) combines a 3D MT integral equation forward

algorithm (Avdeev et al., 1997) with a voxel-based full tensor gravity forward algorithm (Moorkamp et al., 2010). The algorithm solves the non-linear optimization problem using a limited memory quasi-Newton method presented in Avdeeva & Avdeev (2006). Further details on the applied data, inversion scheme, sensitivity-, and model analysis of the congruent resistivity and density models are described in much detail in Franz et al. (2021); Jegen et al. (2016); and Moorkamp et al. (2011) as well as the references therein. Marine MT stations were deployed along two perpendicular profiles along- (profile P100) and across (profile P3) Walvis Ridge, with a station spacing of ~10 km (Fig. 1). Gaps in the final station distribution result from a loss of data or insufficient data quality. The data set is expanded onshore by eight land MT stations from the survey presented in Kapinos et al. (2016). Electrical resistivity and density are sensitive to changes in fluid content (including melt and water), rock density or porosity, as well as rock composition (Bedrosian, 2007; Hinze et al., 2013; Moorkamp, 2022). Therefore, typical breakup related lithologies, such as rifting, magmatic upwelling, and subaerial-, as well as subaquatic phases described in the previous section, should impact the resistivity and density of crustal- and mantle rocks. Since electrical resistivity and density do not necessarily respond likewise to, for example fluid content or geochemical variations, it is reasonable to expect a differentiation of particular resistivity-density pairs identified using a cluster analysis of the congruent resistivity and density models. The determined clusters can then be linked to lithological units and continental break-up stages using geology differentiation (e.g. Bedrosian et al., 2007; Li & Sun 2022; Li et al., 2021; Melo et al., 2017). We therefore present a cluster analysis along the passive margin around Walvis Ridge. Through a comparison to literature parameter values and geophysical studies, we show that the clusters can indeed distinguish lithologies or rock types linked to the continental breakup stages. GMM and other fuzzy clustering methods have also been used to guide joint inversion schemes by adding petrophysical data and improving subsequent geological interpretations (e.g. Astic & Oldenburg, 2019; Astic et al., 2020; Lelièvre et al., 2012; Lösing et al., 2022; Paasche & Tronicke, 2007; and Sun & Li, 2017). For geology differentiation, we assume that the physical parameter models are reliable (Li & Sun, 2022). Therefore, the model uncertainties indicated by sensitivity analysis and model tests presented in Franz et al. (2021) need to be accounted for, when interpreting the clustering results. The tests indicated a model artifact at the northern edge of Walvis Ridge, as well as difficulties to differentiate shallow conductive anomalies related to sediment cover from upper crustal conductive anomalies.

The gravity data inversion (cf. Franz et al., 2021) yields a density anomaly model that defines anomalies from a simple 2-layer reference model containing terrain and Moho topography. We applied the Moho topography correction to account for the strong gravity difference caused by the substantial difference in thickness between the continental- and oceanic crust (cf. Fig. 4 in Franz et al., 2021). Moho depth was derived from a regional density model (Maystrenko et al., 2013) and a global crustal thickness reference (Laske et al., 2013). For consistency, we perform separate cluster analysis for the crustal- and mantle domain according to the applied Moho topography correction. In addition, we isolate a sediment layer for the clustering. We do this, because we observed, that physical parameter clusters show a significant overlap between sediment and upper crustal values and because the foregoing sensitivity tests indicated a difficulty to differentiate between the two

domains. The separation improves the differentiation between the shallowest model features and upper crustal components, increasing the reliability of the geological interpretation (cf. Li & Sun, 2022). The identification of sediment thickness is based on two seismic velocity models of Fromm et al. (2017a) and Planert et al. (2017) along profile P100 and P3, respectively. Since they both use velocity modelling and reflection seismic data to define the sediment basement, the sediment thickness is well constrained. The parameter analysis thus focuses on three domains, which are: a) sediment cover,

b) crust, c) mantle.

In each of these three domains, we perform clustering using logarithmic electrical resistivity ($\log_{10}(\rho_a)$ in Ωm) with absolute density (d in kg m$^{-3}$) of the same model cell. Although the inversion is conducted on a 3D model cube and both data sets have 3D sensitivity, previous tests indicate, that the resistivity model's resolution capabilities are limited away from the MT

stations (Franz et al., 2021). We therefore only use model cells with a maximum horizontal distance of 10 km to the MT station location (Fig. 1). Absolute density values are simply the sum of the inversion density anomaly model, and the crustal-(2810 kg m$^{-3}$) or mantle- (3222 kg m$^{-3}$) density according to whether the model cell belongs to the crustal- or mantle region. The resulting parameter correlations are clustered separately in each of the three domains using the probabilistic Gaussian mixture model (GMM) algorithm (Géron, 2019; McLachlan et al., 2019). GMM generates clusters based on the assumption,

that the model is assembled from a mixture of Gaussian distributions and applies fuzzy- or soft clustering. Such a soft assignment allows for an overlap of clusters, and for each data point (resistivity-density pair) the probability, that the data point belongs to any of the clusters, is determined. We chose the GMM approach, because our two coupled models are smoothed independently from another during the inversion, which leads to model areas of less defined parameter correlations in between the major distinct anomalies. Also, in contrast to hard clustering methods, GMM may dampen

clusters arising from inversion artefacts. For example, a small model artefact may be attributed to an individual cluster, but when analysing the neighbouring cluster's probabilities, the artefact may fall into a different, more geologically plausible cluster. In both cases, the analysis of the probabilities of all clusters helps to evaluate the uncertainty of the clustering and therefore benefits cluster interpretation. Each Gaussian distribution is defined by its mean and covariance matrix, where the covariance matrix defines the shape and orientation of the typically ellipsoid clusters (Géron, 2019). We leave the Gaussian

mixture's covariance matrix unconstrained, to avoid a priori limitations of clusters. In order to achieve the most stable results, we repeated the optimization process 20 times, at which point we experienced no more variation depending on the initial parameter selection in the soft clustering convergence. Due to the initial random cluster selection of the algorithm, classification of the clusters' edges slightly varies within these replicates. While the clusters' centres are robust, the fuzzy overlaps of adjacent clusters experience small changes in the probability estimate, leading to a different primary cluster

assignment. After 20 repetitions, we found that the cluster edges were also robust. The number of clusters in each domain is determined by evaluating the two criteria AIC (Akaike information criteria, Akaike, 1974), and BIC (Bayesian information criteria, Schwarz, 1978), which are theoretical information criteria optimizing data fit (maximum likelihood), while minimizing the number of defined clusters. With both methods, we use the elbow method to determine the minimum

AIC/BIC value with a reasonably small number of clusters. We chose two sediment domain clusters, and each four in the

290 crustal- and mantle domains, respectively (Fig. 2). By correlating the identified clusters with the spatial location along the MT station's profiles, we are able to link them to the geodynamic breakup phases described in the previous section.

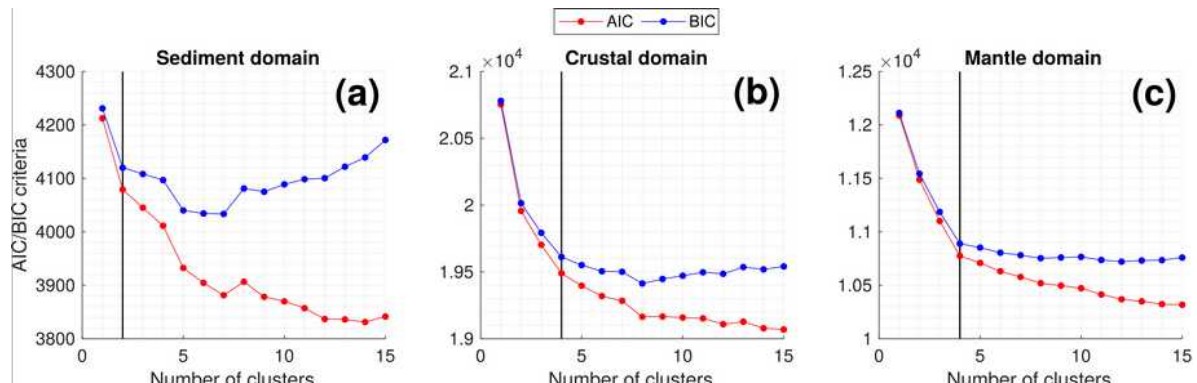

**Figure 2: Theoretical information criteria AIC (Akaike information criteria) and BIC (Bayesian information criteria) to determine the number of clusters in each of the three domains: (a) Sediment domain, (b) Crustal domain, (c) Mantle domain. The vertical**
**black line marks the chosen number of clusters, which is two in the sediment domain, and four in the crustal- and mantle domain, respectively. 4 Results**

*Identified clusters of characteristic physical parameter values and -relationships and their spatial correlation*

Here we describe the electrical resistivity and density clusters for the sediment-, crustal-, and mantle domain and their location within the models along the Namibian margin.

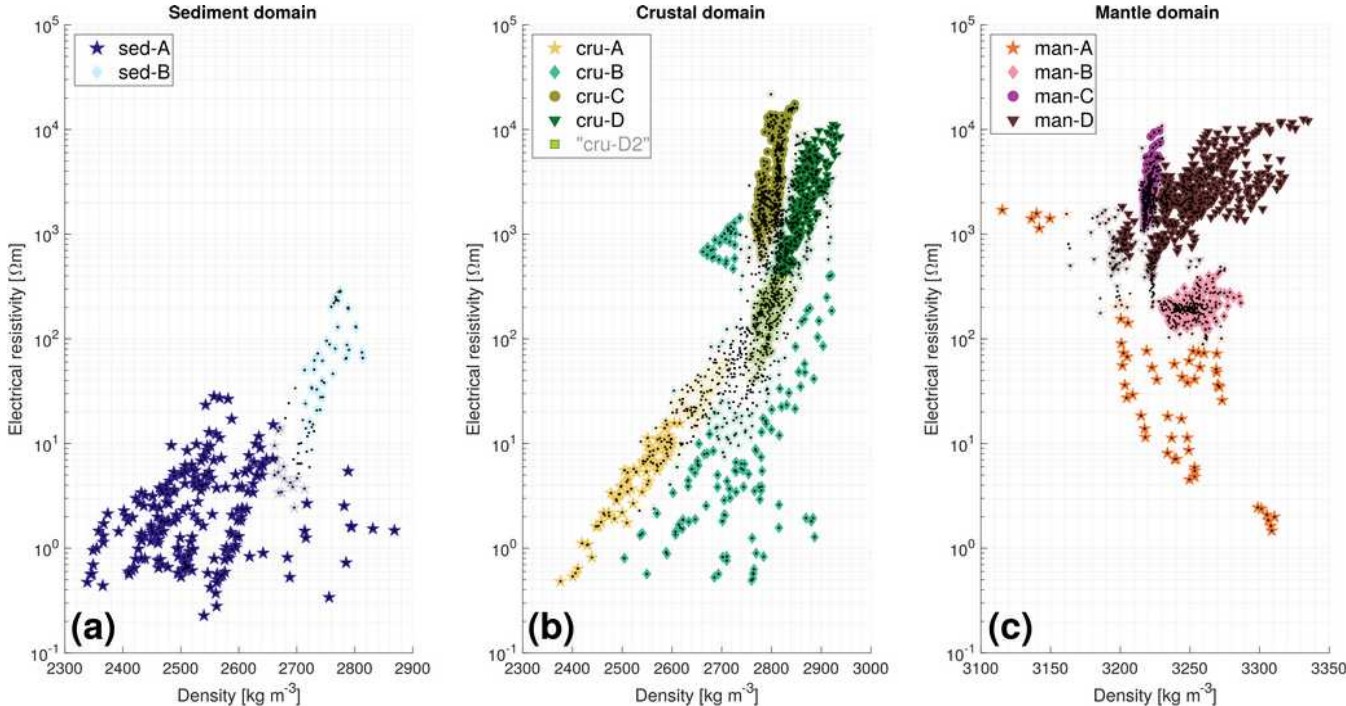

**Figure 3: Cross-plots of electrical resistivity and density and their identified clusters. Colour saturation of symbols represents the posterior probability within Gaussian mixture model: Full saturation of symbols indicates a certainty above 90%, transparent symbols indicate a certainty between 60 and 90%, dots without color saturation indicate a certainty below 60%. (a): Sediment domain (depth extent defined by seismic models of Fromm et al. (2017a) and Planert et al. (2017)). (b): Crustal domain (depth extent defined as sediment basement to Moho derived from Laske et al. (2013) and Maystrenko et al. (2013)). Cluster "cru-D2" is manually defined by dividing the identified cluster cru-D horizontally at 400 Ωm. (c): Mantle domain (below Moho).**

### 4.1 Sediment domain

Clustering of electrical resistivity and density of the sediment domain (as defined by sediment thicknesses retrieved from Fromm et al. (2017a) and Planert et al. (2017)), results in two clusters. In the sediment domain, the evaluated AIC and BIC criteria for choosing the number of clusters is not strictly confined to the number of two. We have additionally evaluated three sediment domain clusters with the GMM algorithm. However, this additional cluster did not result in a third, spatially confined cluster, but only shifted the main two clusters and combined outliers. Therefore, we refrain from incorporating a third cluster, and the two generated clusters have the following characteristics and spatial distribution:

Cluster sed-A (indigo stars in Fig. 3a) comprises low density values (mostly < 2700 kg m$^{-3}$), and low resistivity values (<30 Ωm). The model cells belonging to this cluster are mainly situated in the eastern part of profile P100 (stations MT25 – MT34, Fig. 4) and the central and southern part of profile P3 (MT6 – MT23, Fig. 5). The associated cells cover the entire sediment layer, which in most of these regions has a thickness between 1 and 3 km.

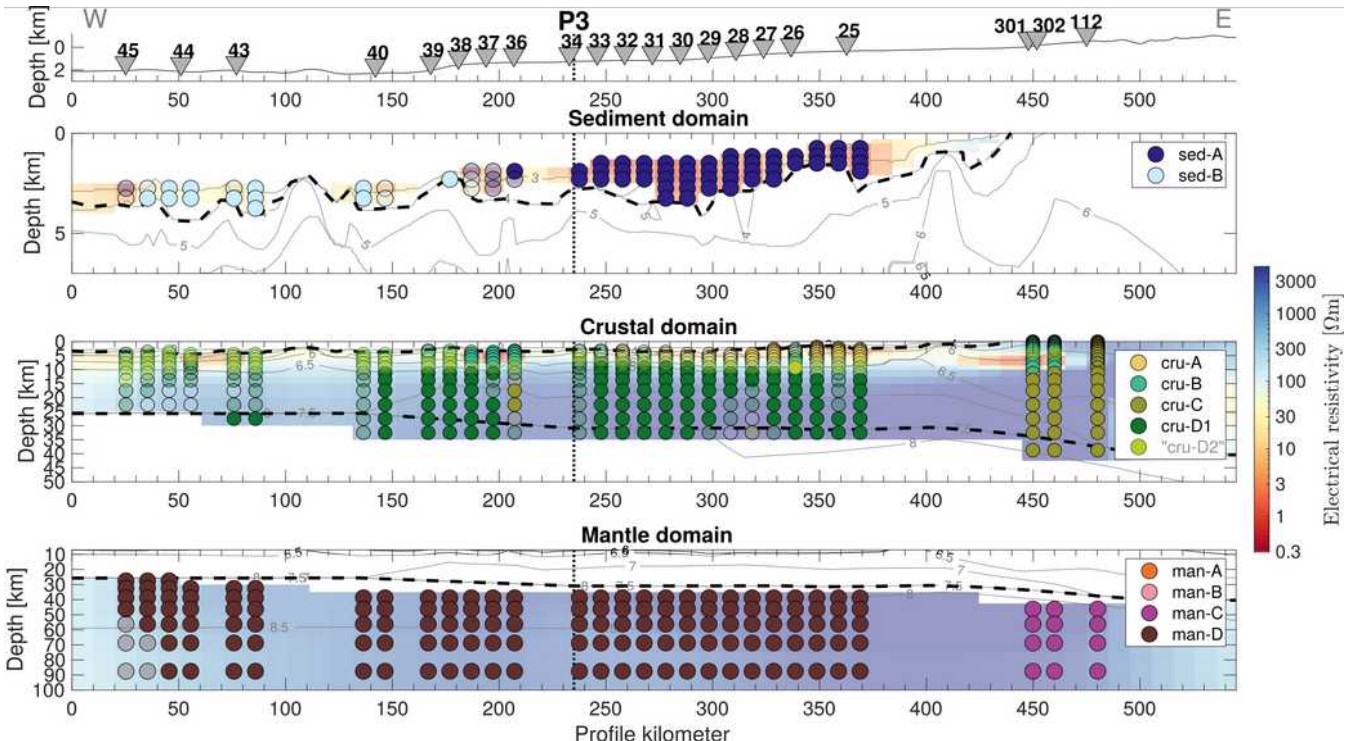

**Figure 4: Vertical section along profile P100 through the 3D electrical resistivity model overlaid with parameter clusters. Top panel shows topography, location of MT stations, and intersection with profile P3. Second panel: Results for sediment domain. Third panel: Results for crustal domain. Fourth panel: Results for mantle domain. Thin lines are seismic velocity contours, thick dashed lines are sediment basement and Moho, (from Fromm et al., 2017a). Vertical dotted line denotes intersection with profile P3. Colour saturation of circles represents the posterior probability of the Gaussian mixture model: Full saturation of circles**
**indicates a certainty above 90%, transparent circles indicate a certainty between 60 and 90%, circles without color saturation indicate a certainty below 60%.**

The second cluster sed-B (cyan diamonds in Fig. 3a) comprises slightly higher densities (> 2700 kg m⁻³) and electrical resistivities (> 10 Ωm, up to ~300 Ωm). The presence of atypically high electrical resistivity values in the sediment cluster B

points to a possible problem with the definition of the sediment domain (see Sect. 3). The sediment thicknesses used to define the sediment domain are from independent seismic velocity models, and may not always exactly coincide with the joint electrical resistivity/density models. However, this may also be due to the vertical model cell size at this depth that is mostly 1 km and thus too large to sufficiently parameterize accurate sediment thicknesses, which leads to a mixed influence of sediments and crust in model cells at the base of the sediment domain. Nevertheless, our investigations have indicated a

lithological difference of the two sediment clusters, which will be further discussed in the next section. In the parameter models, the cells of cluster sed-B are localized further west along profile P100 (stations MT38 – MT45, Fig. 4), and north of Walvis Ridge on profile P3 (stations MT1 and MT2, Fig. 5). Just as we've seen for cluster sed-A, the cells cover the entire sediment domain layer with thicknesses mostly between 1 and 2 km.

Magnetotelluric data resolves, particularly for shallow anomalies, only the conductivity-thickness product, i.e. electrical conductance (Parker, 1980). This can lead to difficulties differentiating between anomalies due to a unit's highly elevated electrical conductivity compared to a unit with increased thickness but only slightly elevated electrical conductivity. To address this ambiguity, we calculate the conductance of each cluster associated with a conductive anomaly (sed-A, sed-B, and cru-A) underneath each marine MT station, by multiplying the sum of the cluster's vertically stacked cells' thickness,

with the stacked cells' mean electrical conductivity. The clusters sed-A and sed-B are clearly separated by their conductance values: sed-A yields values between ~100 and ~5000 S while conductance values of sed-B are much lower with a range between ~40 and ~100 S (Fig. 6).

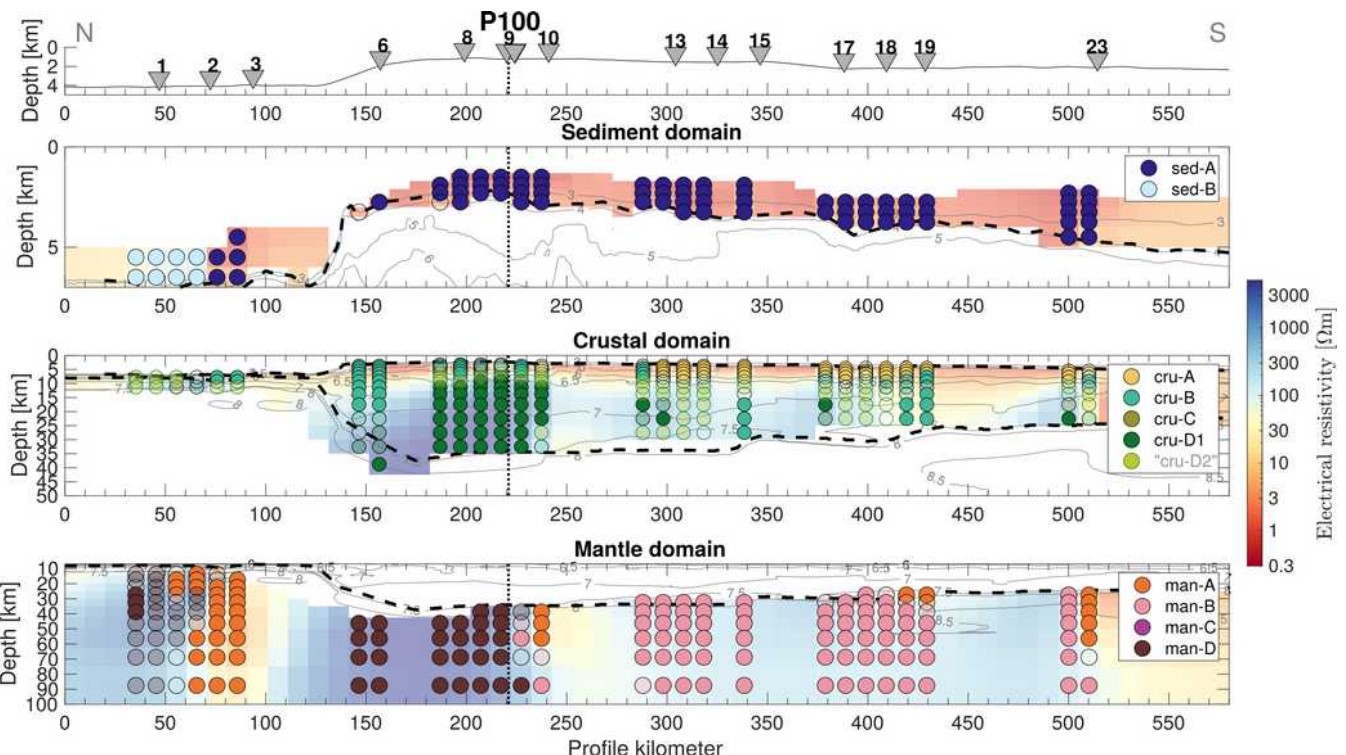

**Figure 5: Vertical section along profile P3 through the 3D electrical resistivity model overlaid with parameter clusters. Top panel**
**shows topography, location of MT stations, and intersection with profile P100. Second panel: Results for sediment domain. Third panel: Results for crustal domain. Fourth panel: Results for mantle domain. Thin lines are seismic velocity contours, thick dashed lines are sediment basement and Moho, (from Planert et al., 2017). Vertical dotted line denotes the intersection with profile P100. Colour saturation of circles represents the posterior probability of the Gaussian mixture model: Full saturation of circles indicates a certainty above 90%, transparent circles indicate a certainty between 60 and 90%, circles without color saturation indicate a**
**certainty below 60%**

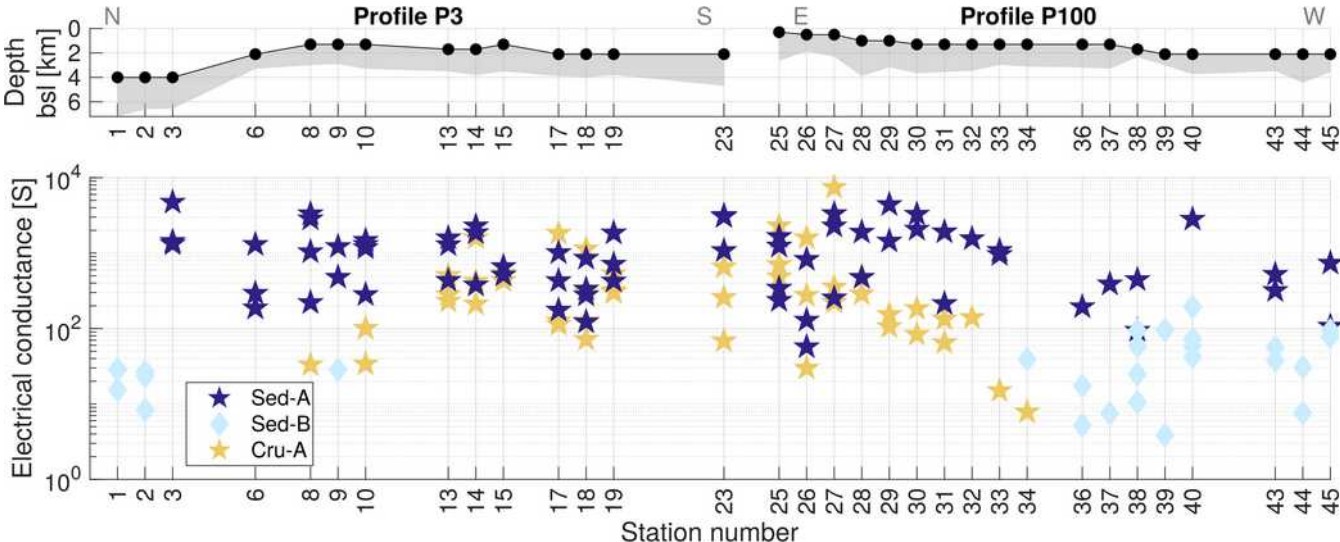

**Figure 6: Depiction of the electrical conductance of three shallow clusters. Top panel shows depth of marine MT stations along profiles P3 (stations 1 to 23) and P100 (stations 25 to 45). Grey area shows thickness of seismically derived sediment thickness (Planert et al., 2017 for profile P3, and Fromm et al., 2017a for profile P100). Bottom panel shows the electrical conductance (conductivity-thickness product) of columns below the corresponding MT station for sediment domain clusters sed-A and sed-B, as well as the crustal cluster cru-A.**

## 4.2 Crustal domain

The crustal domain is defined as the region between the defined sediment base (previous section) and the defined Moho. The Gaussian mixture algorithm places four clusters in this domain (cf. Fig. 2), which describe the following four parameter ranges or relationships.

Cluster cru-A (yellow stars in Fig. 3b) summarizes the low electrical resistivity and low density areas of the model. Electrical resistivity is largely below ~50 $\Omega$m and density below circa 2700 kg m$^{-3}$. The spatial distribution is along the eastern part of profile P100 (stations MT25 – MT32, Fig. 4) and the southern part of profile P3 (stations MT13 – MT23, Fig. 5) and their depth range is less than 9 km below seafloor.

The second cluster cru-B (mint diamonds in Fig. 3b) has the largest variations in terms of parameter ranges and spatial distribution. Electrical resistivity ranges between less than 1 and more than 1000 $\Omega$m, and density values vary from 2500 to 2900 kg m$^{-3}$. The cluster's cells are distributed in all model areas and summarize mostly upper crustal ranges above 10 km (e.g. profile kilometre 180 – 210 on profile P100), but also includes some deep model features (e.g. profile kilometre 140 – 160 on profile P3).

Cluster cru-C (olive circles in Fig. 3b) combines model cells with well-defined densities around 2750 and 2850 kg m$^{-3}$ and
electrical resistivities ranging from ~500 to 20 000 Ωm. The cells are almost exclusively located in the onshore model part and are equally distributed at all depths (0 – 40 km).

The last crustal cluster cru-D (dark green triangles and green squares in Fig. 3b) consists of high resistivity (>100 Ωm), and the highest density (> 2805 kg m$^{-3}$) values. The cluster's cells are distributed over the entire model area. Interestingly, if the
cluster is manually subdivided into a lower and upper resistivity part at 400 Ωm, we obtain a distinct spatial discrimination. Occurrences of cluster "cru-D1", with the high resistivity part, are now constrained to the deeper crust below Walvis Ridge, only. Upper crustal cells along the entire model, and the lower crust south of the ridge are attributed to cluster "cru-D2" (marked as light green squares in Fig. 3b), and exhibit a very similar depth- and spatial distribution as cluster cru-B, with a dominance of cell depths at ~5 – 10 km below seafloor.

**4.3 Mantle domain**

The mantle domain comprises all model cells below the defined Moho, all the way to the model's base at 300 km. Due to decreasing sensitivity, we show the model's sections with marked clusters (Figs. 4 and 5) only to a depth of 100 km, but the parameter cross plot (Fig. 3) includes all depths. The interpretation of our models does not extent below the lithohosphere-asthenosphere-boundary, because the complex mechanisms in the convecting mantle are beyond the resolution capabilities
of our models. This reduced sensitivity to mantle features, and additionally the inversion's smoothing constraint, prohibit precise statements about deep mantle structures. Nevertheless, the parameter clustering helps to distinguish different zones within the upper mantle in the continental margin regime. Our Gaussian mixture modelling places four distinct parameter clusters, characterized by the following aspects:

Cluster man-A (orange stars in Fig. 3c) summarizes mostly cluster outliers, with electrical resistivities ranging from exceptionally low values (1 – 10 Ωm) to ~2000 Ωm and density ranging from 3100 to 3300 kg m$^{-3}$. The associated model cells are mostly located below station MT2, MT3, MT10, and MT23 (Fig. 5). The model areas of these three stations coincides with striking, small-scale low resistivity anomalies across the northern- and southern edge of Walvis Ridge, and the outermost MT station. Model resolution tests for the narrow conductor at ~90 km (coinciding with the Florianopolis
fracture zone north of Walvis Ridge) (Franz et al., 2021), have indicated that a shallow anomaly associated with the fracture zone could be severely smeared due to an effect of sudden change in topography or sediment thickness (Garcia et al., 2015; Worzewski et al., 2012), resulting in the artefacts at mantle depths.

Cluster man-B (light pink diamonds in Fig. 3c) comprises a cluster of low mantle resistivities (between ~100 and ~400 Ωm)
and increased densities of ~3220 to ~3290 kg m$^{-3}$. The cluster's cells are exclusively located at the MT stations south of Walvis Ridge (MT10-23, Fig. 5) and cover all depth ranges of the mantle (25-300 km).

The third mantle domain cluster, man-C (purple circles in Fig. 3c) comprises resistivity values between ~1000 and ~10 000 $\Omega$m and densities around 3220 kg m$^{-3}$, meaning it is very close to the reference value of 3222 kg m$^{-3}$. The associated model cells are distributed at the bottom of the model (> 200 km) along Walvis Ridge (MT25-45), as well as along the entire depth range below the onshore stations (MT100-302).

The last parameter cluster of the mantle domain, cluster man-D (dark red triangles in Fig. 3c) encloses high resistivity (>500 $\Omega$m), and high density (mostly >3220 kg m$^{-3}$) cells, which are located in shallow and medium mantle depths all along Walvis Ridge (stations MT25-44 and MT6-9).

## 5 Discussion

### *Link between the identified physical parameter clusters, passive margin evolutionary phases, and associated crustal types*

By spatially correlating the defined clusters of all three domains, and comparing them to related geophysical models, we have identified a systematic differentiation of the crustal structure from south to north. Thus, we defined three provinces in our survey that are linked to Walvis Ridge, by distinguishing areas south of, along, and north of the prominent bathymetric high. This zoning also reflects the successive south to north unzipping of the South Atlantic opening, and is related to the evolutionary phases described in Sect. 2.

### 5.1 Transitional crust south of Walvis Ridge

The crust south of Walvis Ridge has previously been characterized by an abrupt change from a continental rift zone to a predominantly igneous transitional crust (Bauer et al., 2000; Blaich et al., 2011; Mutter et al., 1988). Extensive sequences of thickened high velocity lower crust, intrusive bodies like dykes and sills, and surface flows in the form of seaward dipping reflectors are representative of this transitional crust (Franke, 2013; Geoffroy, 2005; White & McKenzie, 1989). We attribute the structure inferred from clustering in the southern part of profile P3 (stations MT13 – MT23, Fig. 5) to this series of magmatic activity. It is defined from top to bottom by a) low resistivity (<30 $\Omega$m), high conductance (>100 S), low density (<2700 kg m$^{-3}$) sediments of cluster sed-A; b) very low resistivity (<40 $\Omega$m), high conductance (mostly ≥100 S), low density (<2700 kg m$^{-3}$) upper crust of cluster cru-A, and median resistivity (<400 $\Omega$m) and density (<2850 kg m$^{-3}$) mid- to lower crust of clusters cru-B and cru-D. The cells associated with cru-D south of Walvis Ridge can be mostly attributed to the manually defined cluster "cru-D2", which comprises the cells with electrical resistivity <400 $\Omega$m (see Sect. 4.2). The part of the mantle that is associated with of the southern transitional crust is exclusively defined by c) lower resistivity (<500 $\Omega$m) and median density (3222 – 3300 kg m$^{-3}$) values of cluster man-C.

The low resistivity and density values of cluster sed-A below the southern stations MT13 to MT23 likely represent thick sediments. The electrical conductance values of cluster sed-A (~100-5000 S, Fig. 6) are typical for coastal sediments, where the high values are often indicative of increased sediment thickness, though further mechanisms, such as heat flow, or

sediment type can also impact the electrical conductance (Grayver, 2021). The increased sediment thickness along the post-rift basins of the Namibian margin can be explained by the transport of clastic sediments during the late Cretaceous and early Tertiary (Baby et al., 2018; Brown et al., 2014; Guillocheau et al., 2012; Rouby et al., 2009; Stewart et al., 2000). These sediments may have been transported along the shelf by bottom and surface currents and likely originate from cyclic erosion-deposition phases linked to the Orange, Kuiseb, and Swakop rivers (Dingle, 1992; Dingle et al., 1987; Goslin et al., 1974;

Garzanti et al., 2018). Furthermore, Gholamrezaie et al. (2018) and Maystrenko et al. (2013) observed increased geothermal gradients in the sediment basins along the margin, which are explained with the effect of "thermal blanketing" through the thick insulating cover. Therefore, increased heat flow in these basins can also be regarded as a potential reason for the high electrical conductance values of cluster sed-A.

Below the assumed sediment base (Planert et al., 2017), the upper crust south of Walvis Ridge is characterized by cluster cru-A. Coast perpendicular seismic profiles published and discussed in Gladczenko et al. (1998) and Eldholm et al. (2000) cross our coast parallel profile P3. Slices of our 3D resistivity model that follow these transects are overlaid by seismic line drawings of Gladczenko et al. (1998), and show a correspondence of the upper crustal low resistivity anomalies with occurrences of seaward dipping reflectors (Fig. 7). Analysis of boreholes on the Norwegian shelf (ODP104-642 and

ODP152-917), describe the SDR series as cyclic sequences of alternating layers of massive basalt flows, weathered or vesicular basalt flows, and volcaniclastic, as well as terrigenous sediments and claystones (Eldholm et al., 1987; Larsen et al., 1994; Planke, 1994; Planke et al., 2000). The downhole geophysical data show electrical resistivity values in the range of 5 to 80 $\Omega$m, and densities between 2200 and 2800 kg m$^{-3}$, with the variation being mostly influenced by the proportions of sediments, weathered-, and massive basalts. These proportions are essentially linked to possible continental sediment input,

quantity, and duration of basalt extrusions, as well as the extent of gaps in the extrusion, where surface basalts are subject to weathering.

On the East Faroe High and in the Faroe-Shetland Basin, marine electromagnetic studies have imaged conductive sediments with resistivities in the range of ~5 to 30 $\Omega$m below a massive basalt layer (Heincke et al., 2017; Hoversten et al., 2015;

Jegen et al., 2009; Panzner et al., 2016). The logging data from the Brugdan and Rosebank wells, that are located along the profiles of these surveys reveal respective resistivity and density values of >100 $\Omega$m and 2600 – 2900 kg m$^{-3}$ for the massive basalts, and 2 – 10 $\Omega$m and 2200 – 2500 kg m$^{-3}$ for the underlying sediment sequence. In electromagnetic studies at the Norwegian and Australian passive margins, upper crustal conductors have been interpreted as mineral deposits along igneous intrusions or sills (Corseri et al., 2017; Myer et al., 2013, Spacapan et al., 2020), or graphite precipitation along a detachment

zone (Heinson et al., 2005). These studies also discuss the difficulty to resolve both conductivity and thickness of the thin,

highly conductive mineral layers, and therefore specify electrical conductance values, ranging between $1.5 \cdot 10^4$ and $3 \cdot 10^4$ S. This clearly exceeds the values of our resistivity model (mostly ~50 – 2000 S, Fig. 6), pointing at a different conducting mechanism and thus different lithology.

Consequently, we interpret cluster cru-A to represent an anomaly associated with the seaward dipping reflector series, resulting from an interlayering of clastic sediments and volcanic flows following the interpretation of Bauer et al. (2000); Gladczenko et al. (1998); and Koopmann et al. (2016). The conductance values are largely similar to the range observed in the overlying sediments (Fig. 6), which may be indicative for large amounts of syn-rift sediments or weathered basalts, and few or thin layers of massive basalts, pointing at a periodic extrusive magmatism. The denudation history of the Namibian
margins indicates high sediment deposition rates at the time of continental break-up and directly after, due to dynamic topography and rapid magmatic cooling (Baby et al., 2018; Margirier et al., 2019). The lack of a distinct difference in density between the overlying pure sediment cluster sed-A and the mixed sediment-magmatics cluster cru-A, could be linked to the lack of vertical resolution of the gravity method and a dominant influence of the MT method in the inversion's cross-gradient coupling (Franz et al., 2021) and the increased sensitivity of the MT method for (thin) conductors (Bedrosian,
2007).

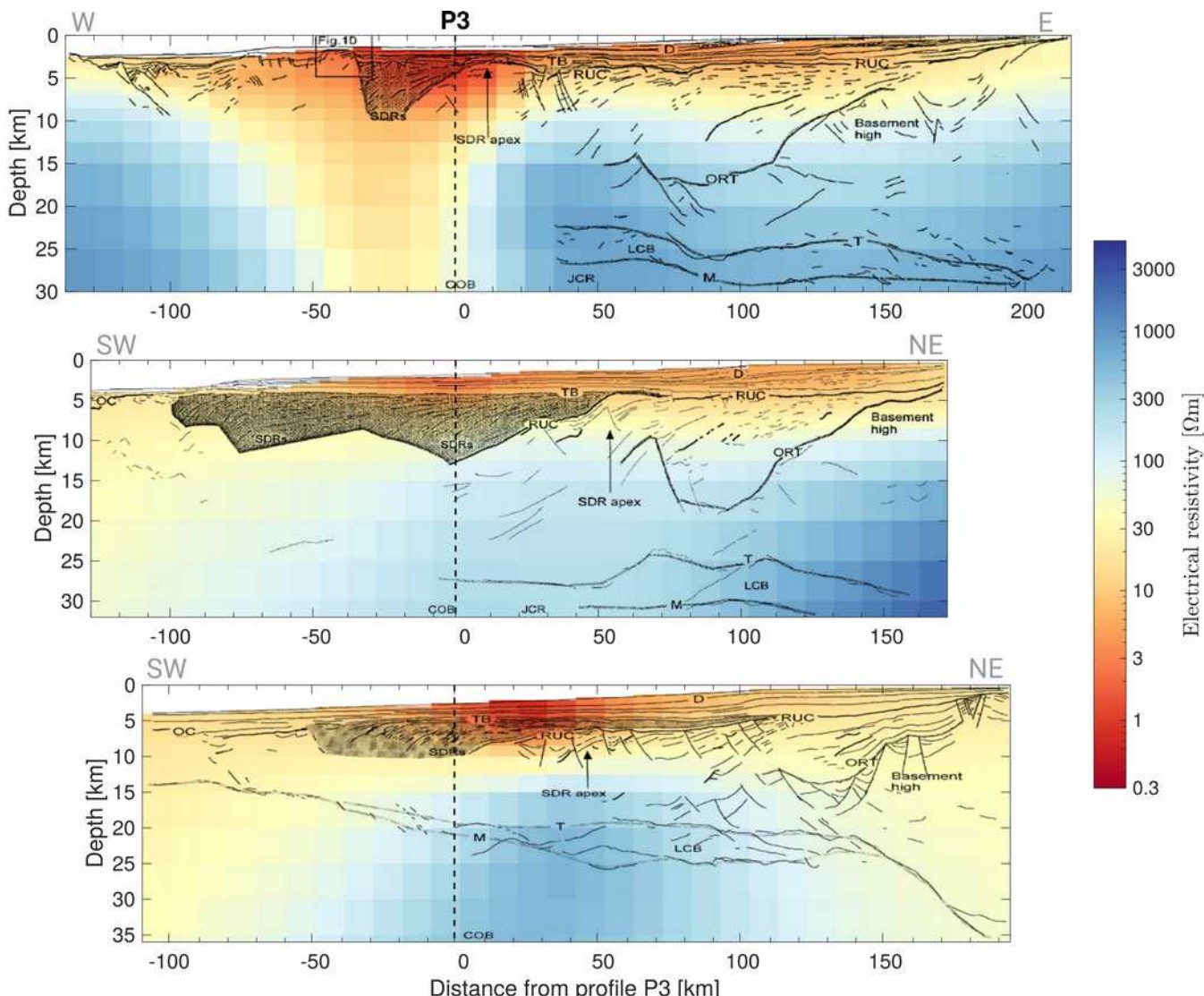

**Figure 7: Vertical slices through the 3D resistivity model along seismic transects presented in Gladczenko et al. (1998) overlaid with their lines drawings of MCS data. From top to bottom are their transects 2, 3, and 4 (their Figs. 4 to 6). Location of transects is shown in Fig. 1. Vertical dashed line is the intersection with our profile P3.**

The middle crust south of Walvis Ridge is assigned to clusters cru-B and cru-D (Fig. 5), where most of the cells associated with cru-D fall into the part of the cluster with electrical resistivity <400 Ωm (referred to as cluster "cru-D2" in Sect. 4.2). The cells associated with cluster cru-B are either situated in the shallow crust or at the transition to conductive features. Therefore, it likely does not characterize a certain lithological unit, but summarizes transitional cells due to model smoothing. The evolution from rifted continental- to oceanic crust at the passive margin is characterized by a growing

imprint of magmatism into the fractured crust and eventually a purely igneous composition (Blaich et al., 2011; White &

McKenzie, 1989). All along the Namibian margin, high mid-crustal velocities (Bauer et al., 2000; Gladczenko et al., 1998; Planert et al., 2017) indicate a mainly igneous composition and the typical rift related structures, for example rotated fault blocks in continental crust, are either lacking (Bauer et al., 2000), or observed closer to the coast (Gladczenko et al., 1998). This indicates a short rifting period with an abrupt continental rupture. At the base of the crust, cumulate bodies and layered
intrusions form magmatic underplating, which has been observed along the entire Namibian margin (Bauer et al., 2003; Blaich et al., 2011; Fromm et al., 2017a; Gladczenko et al., 1998; Hirsch et al., 2009; Planert et al., 2017). Cluster "cru-D2" represents increased values for both electrical resistivity and density (~100 – 400 $\Omega$m, and 2700 – 2850 kg m$^{-3}$) and cells associated with these clusters are distributed in all model parts. We attribute the cluster to the underplated and intruded igneous crust, representing the intrusive counterpart to the extrusive interlayered flows of cru-A.

In the mantle domain, the cluster man-B is remarkably confined, both in the physical parameter space as well as in the spatial extent (Figs. 3 and 5). It clearly clusters all mantle cells of the southern transitional domain. The comparably low resistivity values at the southern transitional domain could indicate a different residual lithospheric mantle compared to the northern, along margin domain. Previous studies have shown a correlation between mantle depletion due to volatile
extraction and increased electrical resistivity (Gardés et al., 2014; Jones et al., 2012; Hirth & Kohlstedt, 1996; Baba et al., 2017). Therefore, we hypothesize the two clusters indicate regions of a less depleted residual lithospheric mantle south of Walvis Ridge, compared to a more depleted residual lithospheric mantle  below the hot spot impingement along the ridge.

## 5.2 Transitional crust along Walvis Ridge

The transitional crust beneath the landfall of Wavis Ridge has likely been directly impacted by the Tristan mantle plume
(Jackson et al., 2000; O'Connor & Duncan, 1990). Therefore, the crust exhibits different characteristics than the southern part, while some mechanisms explained in Sect. 5.1 are similar.

A detailed investigation of the Namibian continental crust's complex composition is not in the scope of this survey, which focuses on the marine environment. Nevertheless, the clusters cru-C and man-C are clearly dominant below the eight
onshore MT stations and thus form clusters characteristic for our model's continental regime. Both clusters stand out with particularly high electrical resistivity values and densities which are very close to the reference (2810 kg m$^{-3}$ for the crustal-, and 3222 kg m$^{-3}$ for the mantle domain). Several seismic studies have thickened crust with high lower crustal velocities (Heit et al., 2015; Ryberg et al., 2022; Ryberg et al., 2015; Yuan et al., 2017),  which have been interpreted to be an effect of magmatic underplating from the thermal anomaly associated with the Tristan mantle plume arrival. In resistivity studies,
cratonic crust is usually expected to have very high electrical resistivities of several thousand $\Omega$m and more (Jones et al., 2009; Moorkamp et al., 2022; Van Zijl, 1977), while mobile belts can show significantly reduced electrical resistivities of ~10 – 50 $\Omega$m (de Beer et al., 1982; Moorkamp et al., 2022; Van Zijl, 1977). The Magnetotelluric studies in the Namibian region have imaged the same behaviour, with very high resistivities in the eastern areas of the Congo craton, and very low

resistivity zones in the Kaoko and Damara belts (Kapinos et al., 2016; Khoza et al., 2013; Ritter et al., 2003; Weckmann et al., 2003). Our electrical resistivity model shows uniformly high resistivity values in the majority of the continental crust, with one near-coast conductive zone between 5-10 km depth (Fig. 4, 410 – 470 km), which could be attributed to one or several of the major shear zone of the Kaoko belt (Goscombe et al., 2003).

Figure 4 shows, that the eastern marine part of profile P100 (~250 – 370 km) is constructed similarly to the southern transitional crust: conductive sediments of cluster sed-A, the conductive upper crust of cluster cru-A that is interpreted as interlayered sediments and volcanic flows, and a middle- and lower crust dominated by cells of cluster cru-D. Directly adjacent to Walvis Ridge, seaward dipping reflectors have been shown to dip to the north instead of the generally westward dipping series along the margin (Elliott et al., 2009; Koopmann et al., 2016). These northward dipping reflectors are attributed to accommodation space that is created by short-lived subsidence at the interaction of the plume and rifting crust, and may explain the development of cru-A type interlayered sediment – magmatic flow packages along Walvis Ridge, which is in accordance with seismic observations by Fromm et al. (2017a). Strikingly, towards the centre of the magmatic activity below Walvis Ridge (MT stations 8 to 10, and 28 to 34) the conductance values of the crustal cluster cru-A decrease (Fig. 6), while the overlying sediments remain at high conductance. We interpret this to depict an increase in the volume of magmatic extrusions, or longer active periods with short breaks. Both result in an increase of massive basalt layers and smaller amounts of conductive sediments or weathered basalts in the seaward dipping reflector series. Another main difference of the transitional crust along Walvis Ridge to that south of the ridge, is the absence of the manually defined cluster "cru-D2". While it describes most middle- to lower crustal cells in the southern domain, the middle- to lower crust along profile P100 is entirely composed of cells attributed to the high resistivity (> 400 Ωm) part of cru-D. Once more, we attribute this to a combination of the longer lasting magmatic overprint due to the halted breakup, and to the impingement of the Tristan plume. Both lead to increased magmatic volumes, including crustal intrusions, magmatic underplating, crustal thickening, as well as uplift. The crustal thickness directly below Walvis Ridge reaches ~33 km compared to ~20 – 25 km in the southern part and the thickness of the magmatic underplating is at ~12 km below Walvis Ridge, and ~4-6 km south of it (Fromm et al., 2017a; Goslin & Sibuet, 1975; Planert et al., 2017).

Corresponding to the increased crustal resistivity below Walvis Ridge, the lithospheric mantle underlying Walvis Ridge is also different to the southern transitional mantle (man-B) with significantly higher resistivity and also density values (Fig. 3c, cluster man-D). Sensitivity tests have shown, that high electrical resistivity values need to reach deep into the mantle, yet absolute values are beyond the resolution capabilities of our study (cf. Franz et al., 2021). We attribute these high lithospheric mantle values to the remnant signature of the upwelling plume center. The emerging of the Tristan plume leads to high ratios of mantle melting and volatile elements are extracted by melts that rise to the surface. This increased magmatism leads to the uplift of Walvis Ridge, thickened crust and rising melts eventually form flood basalts, volcanic flows, and the new oceanic crust (Mutter et al., 1988). The depleted material which is left in the shallow, lithospheric mantle

is highly resistive due to the lack of fluid phases and elements like iron and hydrogen (Baba, 2005; Evans et al., 2005; Matsuno et al., 2010; Selway, 2014).

In contrast to this near-coast part of the profile that portrays the heavily magmatically overprinted crust, the western half of the profile that is located further offshore (Fig. 4, 0 to ~200 km), exhibits a decreasing influence of breakup related magmatism. The crust is still significantly thicker than normal oceanic crust (up to 20 km, Fromm et al., 2017a; Fromm et al., 2017b) demonstrating the impact of the Tristan plume tail, but it lacks the previously discussed typical upper crustal cluster cru-A, which represents interlayered sediments and surface volcanic flows. The absence of these features indicates the transition to submarine spreading and the formation of crust corresponding to pillow basalts, sheeted dykes, and gabbros of oceanic layers 2 and 3 but in a thickened, Icelandic type form (Foulger et al., 2003; Fromm et al., 2017a). Furthermore, the sediment domain reveals a seaward change of the assigned cluster from sed-A to sed-B in the same region, also indicating a transition in sediment regime. The cells of cluster sed-B comprise conductance values which are at least one order of magnitude lower than the conductance values of cluster sed-A (Fig. 6). Although lower conductance is often attributed to sediment thickness (Grayver, 2021), Fig. 6 (top) shows that along both profiles, the seismically imaged sediment thickness deviates repeatedly around circa 2 km. Although slight trends of increasing sediment thickness south- and eastwards are observed, clusters sed-A and sed-B cannot be clearly correlated to sediment thickness. We have previously mentioned, that vertical model cell size at seafloor depth may be too large (mostly 1 km) to accurately parameterize sediment thickness. To evaluate, whether the clear spatial differentiation of the two sediment clusters is biased by the inappropriately large vertical model cell size or conductivity-thickness ambiguities, we resort to a "reverse" conductance calculation. Namely, we use the exact seismically determined sediment thickness at the location of the MT station to estimate electrical resistivity from the model's conductance values. This reverse calculation results in resistivity values of ~0.5 to 10 Ωm for cluster sed-A, and ~10 to 100 Ωm for cluster sed-B, which confirms the cluster analysis results (Fig. 3). We therefore conclude, that the accuracy of the interpolated sediment thickness based on the model's cell thicknesses is sufficient to differentiate between clusters sed-A and sed-B. After eliminating this source of error, we interpret the two clusters' differentiation to mainly result from the sediment layer's electrical resistivity, which indicates a change in sediment type or sea bottom heat flow (Grayver, 2021). The previously described effect of "thermal blanketing" in the southern Namibe basin (Gholamrezaie et al., 2018; Maystrenko et al., 2013), is likely at least partly responsible for the differentiation of the two sediment clusters. This would mean, that the increased resistivity values of cluster sed-B are linked to a decrease in temperature away from the major depositional centre of Walvis Basin. Additionally, a change in sediment type, which could be a decrease in terrigenous input as discussed in Sect. 5.1 as well as a decrease in sediment porosity, can influence the sediment resistivity. Nevertheless, the untypically high absolute sediment resistivity values of cluster sed-B may also be biased by the influence of crustal components due to the inaccurate sediment thickness parameterization related to large vertical model cell size.

### 5.3 Oceanic crust north of Walvis Ridge

Only three MT stations are situated north of the Florianopolis fracture zone (FFZ), of which the closest one (MT03) is likely to experience a strong impact of the drastic topography (cf. Franz et al., 2021). Nevertheless, we aim to discuss the crustal composition by using the other available geophysical models, like the information about Moho depth, as reference, in order to infer the full formation history of the Namibian margin.

The main results are: a) the dominant sediment domain cluster sed-B matches the observation of far offshore sediments along profile P100; and b) the dominant crustal- and mantle clusters cru-D and man-D correlate with the high resistivity-high density clusters observed below Walvis Ridge, even though they appear at much shallower depths. Both of these observations indicate a gradual transition to normal oceanic crust, supporting the hypothesis that a ridge jump at the FFZ transferred breakup related crust to the South American side, making the Angolan coast a rather magma-poor margin (Blaich et al., 2011; Franke, 2013). Sediment domain cluster sed-B (Fig. 5, 0 – 70 km), indicates that the marine sediment cover is similarly thin, as in the far offshore parts on profile P100 (Fig. 4, 0 – 180 km). The thin crust north of Walvis Ridge is mostly represented in cluster cru-D. However, the differentiation in an upper- and lower electrical resistivity cluster, as explained in Sect. 4.2, shows that these cells fall into the lower resistivity cluster "cru-D2". South of Walvis Ridge, we have interpreted this cluster to represent most of the thickened, igneous crust below the extrusive magmatism. However, the observed values are also in accordance with values for normal oceanic crust at magma-poor margins, i.e. relatively high electrical resistivity (several hundred to >1000 Ωm, cf. Heinson, 1999 and Palshin, 1996) and density (~2800 – 2900 kg m$^{-3}$, Carlson & Herrick, 1990; Carlson & Raskin, 1984). The difference to the igneous early-stage breakup crust south of Walvis Ridge, is the significantly reduced crustal thickness (5 – 8 km, Goslin & Sibuet, 1975; Planert et al., 2017). This differentiation puts emphasis on the importance of including all available information (here: seismic sections defining crustal thickness), because the MT-gravity joint inversion alone cannot clearly separate the northern- and southern model parts by parameter values only, as they fall into the same cluster "cru-D2". For the mantle domain, the certainty of cluster classification is significantly reduced, as is illustrated by the decrease in posterior probability of the Gaussian mixture, which is represented by the saturation of symbols in Figs. 3 to 5. The model cells associated with cluster man-D (profile kilometre 39 to 70 of profile P3) generally have low posterior probability values and are close to an association with cluster man-A, which we have classified as a cluster comprising outliers and artefacts. The Florianopolis fracture zone has a very strong influence on the marine MT data, which distorts our model significantly in this area (cf. Franz et al., 2021). Thus, interpretation of the mantle domain in the northern part of our models should be mostly disregarded.

### 6 Conclusion

In this study, the clustering and comparing of the rock parameters density and electrical resistivity lead to the differentiation of distinct crustal types that are associated with the breakup chronology. Additionally, the resulting parameter correlations

may advance joint inversion of MT and gravity data that is based on direct parameter coupling, by providing a starting point for cross-property relationships of electrical resistivity and density in the passive margin regime.

The evaluation and comparison of different geological and geophysical models in the survey area and comparable regions allows several conclusions about the properties of lithological units. However, a few factors indubitably limit the parameter classifications. These limitations include the distribution and resolution capabilities of the two data sets and the inversion method. These limitations comprise the marine MT station distribution along two 2D profiles; the MT method's insufficient

capability to resolve both electrical resistivity and thickness of shallow conductors; decreasing sensitivity with depth; the limited sensitivity of unconstrained gravity inversion for the depth of model anomalies, and thus a joint inversion strongly driven by the fitting of the MT data and minimizing the coupling term; and lastly the presumption of a 1D density background model for the calculation of absolute density values (Fig. 3) and the surfaces chosen to differentiate between the three domains.

Nevertheless, by focusing the parameter clustering on well constrained model areas, using fuzzy clustering of the Gaussian mixture model method to allow cluster overlaps, and discussing the induced uncertainties of the presumptions, we are able to conclude on the following most striking observed features:

1. The differentiation of the sediment domain in a cluster of thick, clastic sediments with increased geothermal gradient, and a second cluster of marine/biogenic sediments. The first (sed-A) occurs mainly at the landfall of Walvis Ridge and south of it and is thus linked to the major depositional centres of the Namibian margin. The second (sed-B) summarizes model areas further away from the shoreline along the ridge and north of it, and represents marine sediments with less terrigenous input.

2. The seismically imaged seaward dipping (SDR) reflectors south of Walvis Ridge show a similar density- and electrical resistivity response as the covering sediment layer. These low density and electrical resistivity values reach up to 9 km into the upper crust, supporting the observation of interlayered sediments and weathered magmatic flows, with either short periods, or lower volume magmatic extrusions. The SDR at the landfall of Walvis Ridge

show decreased electrical conductance, which we interpret to depict an increase in magmatic activity towards the centre of magmatic expulsion below Walvis Ridge.

3. Clustering results in a clear distinction between the response to the magmatic crust south of-, and directly below Walvis Ridge. The crust south of Walvis Ridge is linked to a northward propagating rift and shows a mainly

igneous composition, with layered lower crustal intrusions, and periodic surface volcanic flows (SDR). The crust at the volcanic centre of Walvis Ridge is characterized by accumulation of massive, unweathered magmatics and

uplift, and is thus linked to increased and constant magmatic input, as well as a longer accumulation period due to the halted breakup at the Florianopolis fracture zone.

4. Additionally, the presence of the lower-, and upper electrical resistivity clusters man-B, and man-D suggests a potentially different lithospheric mantle composition. Our hypothesis is, that these variations in the lithospheric mantle composition may result from different degrees of mantle depletion, linked to the increase of volcanic activity at Walvis Ridge due to the crustal impingement of the Tristan plume.

From our results, we propose that Walvis Ridge is the result of a combination of halted spreading and the impingement of the Tristan plume. However, the 3D extent of magmatism that is imaged in our models (which we do not discuss in this article, but in Franz et al., 2021 and Jegen et al., 2016), confirms the observation, that the area of magmatic overprint is too small to justify the arrival of a large plume head (Fromm et al., 2017a; Ryberg et al., 2022). An alternative model explains this smaller scale surface display of the plume by a large, lower mantle upwelling, which tops out at ~1000 km, from where 680  smaller, secondary plumes rise to the surface (Courtillot et al., 2003; French & Romanowicz, 2015; Homrighausen et al., 2019; Zhao, 2007).

With our study, we have defined distinct electrical resistivity – density parameter clusters, linked them with geological units, and compared them to multiple independent geological and geophysical models. The un-biased clustering concept amplifies 685  previous model interpretations, independently. The identification of structures in the MT and gravity data was corroborated by seismic surveys (e.g. by Fromm et al., 2017a; Planert et al., 2017; and Ryberg et al., 2022). This commonality points to the interesting possibility to derive large scale 3D models through acquisition of 3D MT and joint inversion with satellite gravity data and guiding 2D seismic acquisition.

**Author contributions**

GF performed formal analysis, and prepared the original draft. MJ provided resources, engaged in conceptualization, data curation, and in manuscript review and editing. MM implemented software, and engaged in manuscript review and editing. CB provided resources, and engaged in conceptualization, visualization, and manuscript review and editing. WR engaged in manuscript review and editing.

**Competing interests**

The authors declare that they have no conflict of interest.

## Acknowledgements

The Acquisition of the MT data used in this work was supported by the German Research Foundation (DFG) as part of the Priority Program SPP1375 (173329718) and the Future Ocean Program of Kiel Marine Sciences. MM was supported by the German Research Foundation (DFG) under grant 2265/6-1. The authors would like to thank the captain and the crew of R/V Maria S. Merian for the professional and friendly support of the scientific work on the cruises MSM17-1 and MSM17-2. The authors thank our partnering institutes GFZ Potsdam (especially Ute Weckmann, Oliver Ritter, and Magdalena Scheck-Wenderoth) and Alfred Wegener Institute Bremerhaven (especially Tanja Fromm), as well as Lars Planert for their collaboration and the possibility to use their data or models. Further thanks go to Anne Neska, Gerhard Kapinos, and Anna Martí for processing the MT data. We thank Majid Khan, Ulrike Werban and two anonymous reviewers for their constructive comments, which significantly improved this article.

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
