# Peer review of "Formation and geophysical character of transitional crust at the passive continental margin around Walvis Ridge, Namibia"

_EGUsphere, 2022_

## Author Response (AR1)

GEOMAR | Wischhofstr 1-3 | 24148 Kiel | Germany

**Gesa Franz**
**Doctoral Researcher**

Phone +49 431 600-2556
gfranz@geomar.de

+

**Revised manuscript submission**

January 5, 2023

Dear Ulrike Werban, dear Editors,

on behalf of all co-authors I submit the revised manuscript entitled
*"Formation and geophysical character of transitional crust at the passive continental margin around Walvis Ridge, Namibia"* for publication in Solid Earth.

+

We are happy to get the chance to resubmit this manuscript. We have received comments from three referees, which we have incorporated in the revised manuscript.

Additionally, we answered all comments and annotations and, were we did not adjust our manuscript according to the comments, explained our reasons for it. The comments have been uploaded in the public discussion and are added to this letter.

Thank you for accepting to be the editor for this manuscript. Please don't hesitate to contact us, if you have any question.

Kind regards,

Gesa Franz

**GEOMAR**

Helmholtz Centre for
Ocean Research Kiel

Wischhofstr. 1-3
24148 Kiel | Germany

Phone +49 431 600-0
Fax +49 431 600-2805
www.geomar.de

Deutsche Bank AG Kiel
BLZ 210 700 24
Kto. 144 8000

SWIFT/BIC
DEUTDEDB210
IBAN DE
69210700240144800000

Tax Number 2029745781

Sales Tax Code
DE281295378

**Foundation under Public Law**
MinDir Volker Rieke, *Chairman of the Board of Governors*
Prof. Dr. Katja Matthes, *Director* | Frank Spiekermann, *Administrative Director*

[Figure]

**Answer to Anonymous Referee #1** (https://doi.org/10.5194/egusphere-2022-708-RC1):

Thank you for accepting to be a referee, and the time spent to review this manuscript. We appreciate the comments. We have updated our manuscript accordingly and hope to answer the questions satisfactory, in the following. Where we references to manuscript lines, these are the lines in the new, revised manuscript.

*The introduction part is not well-appealing at present, several important references related to passive margin studies and related to geophysical investigation especially electrical resistivity are missing. Authors are advised to present the introduction part and make a connection.*

We found your comment helpful in realizing, that we're missing an introduction to typical geophysical observations at passive margins. Therefore, we added a passage about the typical characteristics seen in geophysical surveys at l. 33 ff. If you have further suggestions for important references, which could benefit our introduction, we would be grateful to have them, so we could include them in our manuscript.

Otherwise, we chose to keep the introduction short, because the second part of the manuscript contains a thorough literature review of the geological setting of the Namibian passive margin. We decided on this form and extensive geological setting, to clearly characterize the margin based on previous research and to be able to reference it in the following discussion of our results. We did not aim to review global passive margins in this manuscript, but focus on the South-East Atlantic margin. In the later discussion, we compare our results to other passive margins (e.g. the Norwegian and Australian margins).

*What's the novelty of this work while already number of inversion techniques have been presented coupled with various modeling schemes?*

The inversion technique and modeling schemes are not a part of this manuscript. They have already been presented in Avdeev et al. (1997), Moorkamp et al. (2010), Moorkamp et al. (2011), Jegen et al. (2016), and Franz et al. (2021).

This manuscript focuses on a more thorough geological discussion of the results. The novelty here is the interpretation of joint inversion results of the Namibian margin. We provide a parameter analysis using a clustering approach of the parameters electrical resistivity and density, which is not available for the Walvis Ridge margin. The new mapping of these two parameters may guide future joint inversion approaches.

*The geology section can be briefly presented rather than defining several phases separately. Authors should focus on depositional history and the relevant tectonic episodes.*

Thank you for this comment. We have added a paragraph about the post-break up depositional history (l. 204 ff.). It is indeed an important part of the geological setting, because we discuss sedimentary processes related to our results in the later part of the manuscript.

The defined geodynamic phases corresponds to relevant tectonic episodes in the formation of the Namibian passive continental margin.

*How does the authors justify the overlap in the electrical resistivity and density plots during clustering analysis in Figure 3? Have the authors considered standardized resistivity-lithology correlation? I guess NO. If yes, how? Also, it is very difficult to distinguish the symbols and colors presented, it can be further simplified for understanding, especially when compare these results to Figure 4.*

The joint inversion algorithm results in regularized, smoothed electrical resistivity and density models. Combined with the applied soft clustering approach, we inevitably experience overlap of the defined clusters (l. 266 ff.). The overlapping values between to domains, e.g. both sediment and crustal domain contain electrical resistivity values between ~0.5 and ~20 $\Omega$m with density between ~2300 and ~2700 kg·m$^{-3}$ are the reason we decided to include the differentiation into three distinct domains. We found it hard to distinguish sediment, crustal, and mantle anomalies without this separation. Li & Sun (2022) have reported similar issues with the reliable identification of parameter relationships associated with the depth of anomalous bodies. We have added this reference to our manuscript, when describing the domain differentiation (l. 247 ff.). The definition of the domains is guided by seismic investigations, which have a good sensitivity for layer boundaries, i.e. sediment basement and acoustic Moho.

Resistivity in itself is not unique. Additionally, the large scale 3D MT inversion does not yield precise absolute electrical resistivity values but aims to interpret large scale variations and localized anomalies. Therefore, a pure resistivity-lithology correlation is inadequate. The main idea of this manuscript is the mapping between two separate geophysical parameters and correlation with tectonic units. Hence, we use the advantages of joint inversion to improve resistivity guided interpretations, which were presented in Jegen et al. (2016).

*The results presented in vertical section along Profile P100 and the cross-plots in Figure 3 are not strongly correlated, there exists overlap in the resistivity values for crustal sediments and others.*

The data presented in figures 3 and 4 is identical. Figure 3 shows a parameter cross plot for all model cells of the 3D models used in the cluster analysis. This includes model cells in a proximity of 10 km to our MT stations (shown in figure 1, cf. l. 259 f). Most of the data points of figure 3 are therefore

represented in either figure 4 or 5, which are the perpendicular profiles P100 and P3 (fig. 1). Therefore, they show the spatial distribution of the defined parameter clusters.

Concerning, the overlap of the parameters in the sediment and crustal domains, please refer to our previous response. The domain separation is specifically included to be able to differentiate between sediment- and crustal anomalies.

**Also, these figures can be better presented.**

We have experimented with different ways to visualize our data and came to the conclusion, that the included figures present our results best. The depiction of all clusters by their physical parameter in figure 3 shows the results of the performed clustering. The two vertical sections in figures 4 and 5 allow for a spatial correlation of the clustering results in order to interpret them in the context of geological variations and match them with the tectonic episodes described in section 2.

If you have additional suggestions to improve presentation, we would appreciate them.

**Again, going back Figure 3, plus, this is too random to get useful conclusions about relationships of two variables.**

Indeed, the initial mapping of electrical resistivity and density of our joint inversion models seemed random. This is the reason why we performed this analysis, using a clustering approach to identify individual groups. This approach prevents personal bias. We additionally separated our models in three depth domains based on seismic results (see previous response concerning parameter overlap) to emphasize geological differentiation. By spatially correlating the clustering results (as shown in figures 4 & 5), we compare the identified clusters to our resistivity and density models, as well as to independent seismic models. From this correlation we conclude our interpretations concerning lithological units and tectonic episodes.

**Why the electrical resistivity and density values are lower in Cluster cru-A? the authors did not justify well. Please justify**

It's true, that this is a surprising results. We interpret the cluster cru-A to represent series of interlayered sediments and basalt flows (seismically imaged as seaward dipping reflectors, SDR). Our model's electrical resistivity and density values are within the bounds of downhole geophysical logs in similar environments (l. 456 ff.). The history of the Namibian passive margin includes several periods of high sedimentation rates, especially during and shortly after continental break-up. These high sedimentation rates can explain, thick sedimentary layers withing the SDR packages. The new paragraph about the

depositional history at the Namibian margin (l. 204 ff.) aids this explanation, and we have also added a sentence referring to this increased sediment input in the discussion (l. 481 ff.).

Another important factor in the low crustal resistivity values, is the MT method's sensitivity to conductors (cf. Bedrosian, 2007), and the deficiency of both applied methods (marine MT and satellite gravity) to resolve thin basaltic layers (l. 483ff.).

***Line 315: The cluster's cells are distributed in all model areas and summarize mostly shallow ranges above 10 km. In this case, 10 Km is not a shallow depth. Also, the authors stated that 189-210 Km on Profile P100, whereas the cross sections don't exceed 100 Km, how would the authors justify this depth contrast and the inferences made?***

We understand that this was incomprehensible.

With the expression "shallow ranges" we refer to the "shallow" or upper crust. The wording was changed to "upper crustal ranges" (l. 372).

The reference to kilometre 180-210 is referring to horizontal profile kilomtre. We added this to the manuscript (l. 373).

***Lines 365-375, I would like to know the ranges defined for low, very low, high and very high etc?***

Thank you for pointing out this missing information. We added the values to our manuscript (l. 431 ff.)

***100 km depth Aur 500 km k profiles using MT Data, are not authentic, please justify.***

Jegen et al. (2016) and Franz et al. (2021) discuss the resolution and uncertainty of the electrical resistivity models. Synthetic tests have verified, that the vertical extent of anomalies may not always be reproduced accurately, i.e. shallow conductors may be vertically smoothed. This may be explained by large vertical cell size, and in some model areas by insufficient station spacing. Additionally, we may not differentiate between high absolute resistivity values at depth, i.e. a change in resistivity at Moho depth. Nevertheless, the synthetic tests have proved the need for generally high resistivities (>2000 $\Omega$m) at lower crustal and upper mantle depth (clusters cru-D and man-D), and the need for upper crustal conductors identified by cluster cru-A. Thus, our model tests proved a reasonable sensitivity for the most important model features, which we aim to interpret.

Although our clustering includes all inversion model cells (up to 300 km depth), we have confined our interpretation mostly to the upper 100 km (shown in figures 4 and 5), due to reduced sensitivity below. The models reach this deep to prevent boundary influences from the 1D background model needed for inversion (cf. Franz et al., 2021).

*And there is no relationship between conductance and density (it should be a cross verification while resistivity to density relations is defined)?*

Magnetotelluric data are, particularly for shallow conductors, mostly sensitive to electrical conductance, i.e. the conductivity-thickness product. When interpreting conductivity anomalies, we are thus faced with ambiguity. By calculating the electrical conductance, i.e. multiplying the cell's conductivity values with the cell's thickness, we can verify the distinction between the two sediment clusters. The analysis emphasizes, that the distinction is not only an effect of sediment thickness, as the two clusters also clearly vary in conductance. We are therefore confident, that the two clusters summarize different lithological units (cf. l. 337 - 345).

Additionally, we evaluate whether the effect is biased by an inaccurate discretization of sediment thickness due to a large vertical cell size. For this, we "revert" the conductance calculation by dividing conductance by precise sediment thicknesses derived from seismic data. The resulting electrical conductivity values are in the range of the model's values (cf. l. 580 – 586).

*Figure 7, I don't understand the smallest values (0.3-3 Ohm-m) in a depth range of 0-10 Km (even onwards). Can the author justify such an anomaly ?*

We are similarly surprised by these results. Please refer to our previous answer to the question *"Why the electrical resistivity and density values are lower in Cluster cru-A?"*. In summary, we believe that this conductive anomaly in fact images packages of interlayered thick sediments and volcanic flows.

The overlay with the seismic sections by Gladczenko et al. (1998) in figure 7 emphasizes the immense depth (10 km and more) of the seismically imaged SDR sequences. These thicknesses have been confirmed by various authors, e.g. Bauer et al. (2000); Elliott et al. (2009); and Koopmann et al. (2016).

*Also, there are several grammatical mistakes, and some sentences are very difficult to understand, please do a complete overhauling for the English write up.*

We kindly asked a native English speaker colleague to help us revise the manuscript.

**References in this response:**

Avdeev, D. B., Kuvshinov, A. V., Pankratov, O. V., and Newman, G. A.: High-Performance Three-Dimensional Electromagnetic Modelling Using Modified Neumann Series. Wide-Band Numerical Solution and Examples, Journal of geomagnetism and geoelectricity, 49, 1519–1539, 10.5636/jgg.49.1519, 1997.

Bauer, K., Neben, S., Schreckenberger, B., Emmermann, R., Hinz, K., Fechner, N., Gohl, K., Schulze, A., Trumbull, R. B., and Weber, K.: Deep structure of the Namibia continental margin as derived from integrated geophysical studies, Journal of Geophysical Research: Solid Earth, 105, 25 829–25 853, 10.1029/2000JB900227, 2000.

Bedrosian, P. A.: MT+, integrating magnetotellurics to determine earth structure, physical state, and processes, Surveys in Geophysics, 28, 121–167, 10.1007/s10712-007-9019-6, 2007.

Elliott, G. M., Berndt, C., and Parson, L. M.: The SW African volcanic rifted margin and the initiation of the Walvis Ridge, South Atlantic, Marine Geophysical Researches, 30, 207–214, 10.1007/s11001-009-9077-x, 2009.

Franz, G., Moorkamp, M., Jegen, M., Berndt, C., and Rabbel, W.: Comparison of Different Coupling Methods for Joint Inversion of Geophysical Data: A Case Study for the Namibian Continental Margin, Journal of Geophysical Research: Solid Earth, 126, 1–28, 10.1029/2021jb022092, 2021.

Gladczenko, T. P., Skogseid, J., and Eldhom, O.: Namibia volcanic margin, Marine Geophysical Researches, 20, 313–341, 10.1023/A:1004746101320, 1998.

Jegen, M., Avdeeva, A., Berndt, C., Franz, G., Heincke, B., Hölz, S., Neska, A., Marti, A., Planert, L., Chen, J., Kopp, H., Baba, K., Ritter, O., Weckmann, U., Meqbel, N., and Behrmann, J.: 3-D magnetotelluric image of offshore magmatism at the Walvis Ridge and rift basin, Tectonophysics, 683, 98–108, 10.1016/j.tecto.2016.06.016, 2016.

Koopmann, H., Schreckenberger, B., Franke, D., Becker, K., and Schnabel, M.: The late rifting phase and continental break-up of the southern South Atlantic: the mode and timing of volcanic rifting and formation of earliest oceanic crust, Geological Society, London, Special Publications, 420, 315–340, 10.1144/SP420.2, 2016.

Li, X. and Sun, J.: Towards a better understanding of the recoverability of physical property relationships from geophysical inversions of multiple potential-field data sets, Geophysical Journal International, 230, 1489–1507, 10.1093/gji/ggac130, 2022.

Moorkamp, M., Jegen, M., Roberts, A., and Hobbs, R.: Massively parallel forward modeling of scalar and tensor gravimetry data, Computers & Geosciences, 36, 680–686, 10.1016/j.cageo.2009.09.018, 2010.

Moorkamp, M., Heincke, B., Jegen, M., Roberts, A. W., and Hobbs, R. W.: A framework for 3-D joint inversion of MT, gravity and seismic refraction data, Geophysical Journal International, 184, 477–493, 10.1111/j.1365-246X.2010.04856.x, 2011.

**Answer to Anonymous Referee #2** (https://doi.org/10.5194/egusphere-2022-708-RC2):

Thank you for accepting to be a referee, and the time spent to review this manuscript. We appreciate the comments. We have updated our manuscript accordingly and hope to answer the questions satisfactory, in the following. Where we references to manuscript lines, these are the lines in the new, revised manuscript.

**The main comments are how to evaluate the uncertainty of your clustering? Where do the uncertainties come from, and how large? For example, in Figure 4 (second panel), the whole area was identified as sed-A within profile 250-350 km for all depths, it is not clear if MT method/the resistivity does not have enough resolution to identify the resistivity change along the depth axis?**

The uncertainty of the clustering is determined within the probabilistic Gaussian mixture modeling algorithm. GMM is a soft clustering approach, meaning that for each data point the probability for each defined cluster is determined and then the cluster with the highest probability is assigned to the data point (cf. l. 266ff.). We visualized the probability of the assigned cluster with a colour saturation in figures 3, 4, and 5. We slightly modified the figures, by including only three ranges of certainty: above 90%, between 60 and 90%, and below 60%, which are thus better distinguishable by the symbol saturation in figures 3, 4, and 5. We believe, this improves the direct visibility of model areas with larger clustering uncertainty. For example in figure 3b at the intersection of clusters at density 2600-2800 kg m$^{-3}$ and el. Resistivity $2 \cdot 10^1$ - $10^2$ Ωm; or in figure 5, bottom panel, profile kilometre 30 – 70. For this second example of the mantle cluster assignment along profile P3, we discuss the cluster uncertainty in l. 620ff., stating that interpretations in the northern part of our models should be mostly disregarded.

To clearly state that the posterior probability is a measure for uncertainty, we changed the sentence in l. 271f.

In the following figure (Fig. 1 in this reply), we have also depicted the distribution of the probability for each of the 11 clusters. The distribution shows mostly a dominant high certainty (<90%) except for the manually defined cluster "cru-D2". In our opinion, this only emphasizes the importance to isolate this cluster manually and distinguish it from the more clearly defined cluster cru-D, representing magmatic underplating below Walvis Ridge.

[Figure]

*Figure 1: Representation of cluster certainty. It shows the distribution of the Gaussian mixture model's posterior probability for each cluster. Binning is fixed between the minimum in all probabilities (35%) and 100% in steps of 5%.*

Concerning cluster sed-A and the lack of vertical differentiation: In Figure 2, we present the AIC and BIC criteria for the determination of the number of needed clusters. For the sediment domain (Fig. 2a), the curve is generally less smooth than for the other two domains, which complicates the decision process for the number of clusters. Therefore, we also tested clustering with three assigned sediment domain clusters. The additional third cluster mainly moved the transitional zone between the lower and higher resistivity clusters and did not add a spatially confined, thus lithologically interpretable cluster (see Fig. 2A and 3A in this reply). Additionally, the certainty, i.e. posterior probability of the assigned clusters, decreased with the addition of a third cluster (see Fig. 2A in this reply, less saturated symbols).

In another attempt, which we did not include in the final manuscript, we have added seismic velocity as a third parameter in the clustering algorithm (Fig. 2B, 2C, and 3B in this reply). We used the available velocity models by Fromm et al. (2017) and Planert et al. (2017). The inclusion of seismic velocity in the clustering with three sediment domain clusters resulted in a clear vertical separation of the lower resistivity sediment cluster (Fig. 2B and 3B in this reply). We interpreted this differentiation do result from the increasing compression with depth, and therefore increasing seismic velocity of marine sediments. Figure 2B in this reply shows that the electrical resistivity and density values of these vertically separated clusters (here, sed-A and sed-B) are in the same ranges. This indicates that the resolution of our resistivity and density models are insufficient to resolve the parameter change due to increasing compression. Nevertheless, for our differentiation in lithological or tectonic units, this effect of mere compaction is insignificant, which is why we have refrained from including it in the final manuscript. For the Crustal- and Mantle domains, this effect was even stronger, which can be attributed

to the general structure of seismic velocity models, which are usually built as gradient models with strictly increasing velocity with depth.

We added a brief statement about the effects of the inclusion of a third sediment domain cluster in our manuscript (l. 307ff.).

We hope this explanation of our additional clustering tests helps to understand our reasoning for the two evaluated sediment clusters in the final manuscript, and the neglecting of the three parameter clustering including seismic velocity.

[Figure]

*Figure 2: Cross-plots of electrical resistivity and density anomaly (A and B), and electrical resistivity and seismic velocity (C) and their identified clusters for the sediment domain. The analyses generated three sediment domain clusters, compared to the 2 clusters presented in the final manuscript. Additionally, plots B and C show the results for a clustering of the three parameters electrical resistivity, density, and seismic velocity. Note that in this figure, density is depicted as an anomaly of a background density of 2800 kg m$^{-3}$. For absolute density as it is depicted in the final manuscript, one simply adds those 2800. These results were not included in the final manuscript and therefore only serve as an explanation for our reasoning to not include them.*

[Figure]

*Figure 3:* *Vertical section along profile P100 through the 3D electrical resistivity model overlaid with parameter clusters. Top panel shows topography, and location of MT stations. Second panel (A): Results for sediment domain and two parameter (electrical resistivity and density) clustering. Third panel (B): Results for sediment domain and three parameter (electrical resistivity, density, and seismic velocity) clustering. Colour saturation of circles is proportional to posterior probability of the Gaussian mixture model. Less saturated circles depict decreased certainty of the cluster classification. These results were not included in the final manuscript and therefore only serve as an explanation for our reasoning to not include them.*

**In Figure 6, how does the electrical conductance distribute for other clusters?**

We have only included electrical conductance values for the shown three clusters, because they are the clusters summarizing the upper model conductive anomalies above 10 km. In MT inversion, it is often difficult to resolve both parameters electrical conductivity and thickness for shallow layers, whereas the total conductance of a layer may be resolved with relatively high accuracy (Kaufmann et al., 2014; Weidelt, 1985). This is due to the limited high frequency signal, especially in marine MT studies where the high frequency content of the source signal is filtered by the water column. For the deeper and more resistive structures, we assume that we can ambiguity concerning electrical resistivity and thickness of anomalies is less pronounced.

**Page 1, Line 27-30, About the end-member of passive margins, the volcanic or non-volcanic, or called magma-poor or magma-rich margins, Authors need to mention a form between the volcanic and non-volcanic margins, an intermediary form margin in the world.**

Thank you, for pointing out this lacking information. We have added a sentence and the proposed references to our manuscript (l. 32 ff.) and clarified, that the described magma-poor and magma-dominated margins are only the end members of margin classification (l. 28).

**Page 2, Line 36-37, it requires reference here to explain "margin formation and mantle plume-lithosphere interaction."**

We have added five exemplary references (l. 53f.).

**Page 2, Line 57, the title "Geological Setting: Phases of the geodynamic evolution of the Namibian passive continental margin" is too long and it requires delete the redundant words after geological setting.**

We have followed the referee's suggestion for the Results-part title and included the *"Phases of the geodynamic evolution of the Namibian passive continental margin"* as a subtitle. We believe, that the descriptive subtitles help guiding the readers.

**Page 5, Line 100-103, needs to add reference here**

We have added four references connecting decompression melting with rifting lithosphere (l. 125 f.). References for the features a) intrusive magmatic bodies, b) magmatic dykes and sills, and c) continental flood basalts (CFB), are included in the following paragraphs, where we present examples for each of the features.

**Page 5, In this section "Phase 2: Arrival of magmatism", Authors need to add Figure 1 after some text to show the "Kaoko Belt" or intrusive bodies, dyke and sills, and the location of "Tristan mantle plume".**

We added the reference to Figure 1 (l. 139) and altered the map inlay of Figure 1 to show the present day location of the Tristan hotspot and the corresponding hotspot track.

**Page 5, Line 123 "2.3 Phase 3: Transition from rifting to continental breakup", I think this title is questionable, why authors separate the rifting process from continental breakup? This process is not continuous?**

Yes, we agree, that the transition from rifting to continental break-up is a continuous process.

We have observed that in literature, authors often describe a stringent COB (continent-ocean boundary), which is presented as a line feature. As mentioned in l. 150ff., we believe that this distinct definition of a boundary is prone to false interpretations, and the transition should be rather viewed as a zone. This zone may present differently at different margins, but similar processes are responsible for the associated structures. Therefore, we compiled a specific sub-section describing this "transition from rift to drift". This phrase has been used similarly by e.g. Bertotti et al. (1993); Jagoutz et al. (2007); and Nielsen and Hopper (2004).

We changed the second sentence of this paragraph, to clearly state that the rift to drift transition is a continuous process (l. 151).

**Page 8, Line 196, please simply describe the gravity inversion method and tools used in this manuscript.**

We have added basic information about the applied inversion code and forward engines in l. 215ff. Additionally, we added a reference at the proposed line to the previous publication Franz et al. (2021), where the technical details of this joint inversion are described in much detail (l. 238).

We refrained from describing the joint inversion details here, because it has been covered extensively in the mentioned previous publication.

**Page 8, Line 222-226, the method discussion needs move to discussion part.**

We believe, that this part is important as a motivation why we utilized the GMM clustering method. We do not discuss our particular clustering here, but describe the general benefits of this approach.

In in discussion in l. 620ff. we pick up on this benefit: the clustering results north of Walvis Ridge have generally lower posterior probability. There, the soft clustering approach helps to identify areas of less certainty. Consequently, we mostly refrain from interpreting anomalies there (l. 623f.).

**Page 10, the title "Results: Identified clusters of characteristic physical parameter values and -relationships and their spatial correlation" is too long. The title "Identified clusters of characteristic physical parameter values and -relationships and their spatial correlation" can be as a sub-title.**

We have followed the referee's suggestion and included the *"Identified clusters of characteristic physical parameter values and -relationships and their spatial correlation"* as a subtitle. We believe, that the descriptive subtitles help guiding the readers.

**Page 15, Line 357, the title "Discussion...." is also too long.**

We have followed the referee's suggestion for the Results-part title and included the *"Link between the identified physical parameter clusters, passive margin evolutionary phases, and associated crustal types"* as a subtitle. We believe, that the descriptive subtitles help guiding the readers.

**Page 23, the conclusion part, Line 608-613, please remove to the discussion part.**

This paragraph summarizes the work we presented in this manuscript, justifies our results by mentioning the agreement with independent seismic results, and states how these results can be useful for future research. It follows the discussion and we would like to leave it in the manuscript as a concluding statement.

**Figures**

**In Figure 1, add the location of the "Tristan mantle plume"**

We altered the map inlay of Figure 1 to show the present day location of the Tristan hotspot and the corresponding hotspot track. We do not display the supposed location during break-up at the Namibian margin, because it is subject to discussion (cf. l. 670ff.).

**In Figure 2, add the letter A, B and C for separated small figures. The labeled AIC/BIC mark above the three small figures.**

Figure two was changed according to the referee's suggestions.

**References in this response:**

Bertotti, G., Picotti, V., Bernoulli, D., and Castellarin, A.: From rifting to drifting: tectonic evolution of the South-Alpine upper crust from the Triassic to the Early Cretaceous, Sedimentary Geology, 86, 53–76, 10.1016/0037-0738(93)90133-P, 1993.

Franz, G., Moorkamp, M., Jegen, M., Berndt, C., and Rabbel, W.: Comparison of Different Coupling Methods for Joint Inversion of Geophysical Data: A Case Study for the Namibian Continental Margin, Journal of Geophysical Research: Solid Earth, 126, 1–28, 10.1029/2021jb022092, 2021.

Fromm, T., Jokat, W., Ryberg, T., Behrmann, J. H., Haberland, C., and Weber, M.: The onset of Walvis Ridge: Plume influence at the continental margin, Tectonophysics, 716, 90–107, 10.1016/j.tecto.2017.03.011, 2017.

Jagoutz, O., Müntener, O., Manatschal, G., Rubatto, D., Péron-Pinvidic, G., Turrin, B. D., and Villa, I. M.: The rift-to-drift transition in the North Atlantic: A stuttering start of the MORB machine?, Geology, 35, 1087, 10.1130/G23613A.1, 2007.

Kaufman, A., Alekseev, D., and Oristaglio, M.: Principles of Magnetotellurics, pp. 377–415, 10.1016/B978-0-444-53829-1.00011-3, 2014.

Nielsen, T. K. and Hopper, J. R.: From rift to drift: Mantle melting during continental breakup, Geochemistry, Geophysics, Geosystems, 5, 10.1029/2003GC000662, 2004.

Planert, L., Behrmann, J., Jokat, W., Fromm, T., Ryberg, T., Weber, M., and Haberland, C.: The wide-angle seismic image of a complex rifted margin, offshore North Namibia: Implications for the tectonics of continental breakup, Tectonophysics, 716, 130–148, 10.1016/j.tecto.2016.06.024, 2017.

Weidelt, P.: Construction of conductance bounds from magnetotelluric impedances., Journal of Geophysics, 57, 191–206, 1985.

**Answer to Anonymous Referee #3** (https://doi.org/10.5194/egusphere-2022-708-RC3):

Thank you for accepting to be a referee, and the time spent to review this manuscript. We appreciate the comments. We have updated our manuscript accordingly and hope to answer the questions satisfactory, in the following. Where we references to manuscript lines, these are the lines in the new, revised manuscript.

**Larger Issues, Point 1: Handling of errors.**
Handling of errors of the clustering have been discussed in the previous answer to referee #2.

As I understand, this issue concerns the errors of the models themselves. The electrical resistivity and density model errors are difficult to derive due to the ambiguity of geophysical inversion. Therefore, it's not a common practice to specify model errors. Usually model anomalies are tested for their necessity, which we performed in the previous publication (Franz et al., 2021). Here, the analysis of models is focused on comparing differences in the models' parameters along the passive margin. The clustering algorithm is used to identify zones of common parameter relationships and distinguishing from zones with different parameter relationships. This zoning is then linked to geological processes to enhance passive margin interpretation.

**Larger Issues, Point 2: Purpose of manuscript.**
Our referenced previous publication is a discussion of different approaches for the joint inversion data and model integration. We thoroughly describe models and data, joint inversion technical aspects, and discuss model resolution and uncertainties. It includes a short discussion about the geological interpretation of the results, which is only based on the physical models and comparison to previous research.

This manuscript presents user-unbiased approach to model interpretation, and a more thorough geological discussion based on the Namibian margin's history. Clustering of the model parameters is a significant improvement to model interpretation, because it offers independent assignment of different model areas. These model areas are then linked to lithological units and validate the previous subjective interpretations.

We have added short explanations of the benefits of the clustering concept to the introduction (l. 67 f.) and conclusion (l. 679 f.).

**Larger Issues, Point 3: Hypothesis of different magma sources south of – and below Walvis Ridge**
In our manuscript we state the hypothesis, that the crustal structure south of Walvis Ridge and along the ridge differ as a result of the direct plume impact at Walvis Ridge latitudes. We note, that the involvement of the Tristan plume is a topic to debate (l. 99 f.). This comment helped us to realize inconsistencies in our hypothesis. What we actually wanted to state, is a difference in the crust and

upper mantle related to the degree of mantle and melt depletion. And to link the higher depletion below Walvis Ridge to the impingement of the Tristan hot spot and the corresponding extraction of volatiles. The residual upper mantle would then be more depleted compared to the hypothesized "rift related" crust south of Walvis Ridge. We have rephrased the parts of the discussion and conclusion to clearly describe this hypothesis (l. 509ff. and 665 ff.).

We are not specialists in isotope geochemistry, but have evaluated the proposed papers. We believe that their conclusions do not contradict our statements. We state that the earliest phase of continent break-up is associated with rifting and that that early magmatics are mainly of upper mantle composition (in l. 100 f.). Gibson et al. (2006) also link this earliest stage of the CFB emplacement (~145 Ma) to melts at the mechanical boundary layer (MBL) at ~150 km depth and not a deep plume source. We do not rule out involvement of plume material in the Etendeka CFB in the subsequent stages. In fact we point out the interaction of the Tristan plume and the lithosphere and heterogeneous composition (l. 131 ff.) of intrusive magmatics, which we link to the ascend of magma which forms dykes and eventually the CFB (l. 136, 142).

Concerning the comment about the model depth and depth of depleted mantle: Thank you for describing this problem of a mismatch of mantle convection and our statements about a different mantle structure south of-, and along Walvis Ridge. We understand that there needs to be clarification, because we haven't clearly stated that interpretations should be confined to the upper/lithospheric mantle only. We added appropriate statements: We point out, that the resolution capabilities of the electrical resistivity model decrease with depth, and the statements therefore become more vague with depth (l. 635 f.). Additionally, we clearly phrased that interpretations of the mantle domain should not extent below the LAB in l. 389 ff. In our discussion of the mantle clusters, we also added the explicit statements, that our interpretations concern the shallow, lithospheric mantle (l. 513 f. and 562 f.).

**Larger Issues, Point 4: Salt north of Walvis Ridge**
Salt deposits north of Walvis Ridge have been mapped offshore Angola in the Kwanza basin north of ~15°S (e.g. Blaich et al., 2011; Moulin et al., 2010; Strozyk et al., 2017, Torsvik et al., 2009). The salt directly adjacent to the FFZ may have been sheared off to the South American margin during the Albian ridge jump. The latitudes north of 15°S are not included in our model area. Therefore, we do not discuss any inclusion of salt horizons in our model region.

**Direct comments:**
**Line 35: mantle**
Corrected.

**Line 62: check ages of rifting**
we rephrased the sentence slightly and included short time spans to indicate the uncertainty of different plate reconstructions stating different times (l. 82 f.)

**Line 71: basement**
Changed "background" to basement (l. 92).

**Line 79: technically this is volcanic. There are no constraints as to whether plutonic rocks were generated initially. The timing of magmatisim and rifting is of considerable controversy - please examine the following papers and incorporate their insights. As you will see from the papers, the correlation between magmatism and rifting is not quite as portrayed.**
We added the explanation of the alternative hypothesis of an early plume impact and major influence in rift initiation (l. 106 ff.).
Generally, this phase summarized the very early rift phase and we only discuss the potential triggers for this rifting. We have now included both hypotheses: "plume inducing", and "plume induced". We describe the arrival of magmatism in the second phase, where the intrusive and extrusive magamtics are described in more detail. To avoid confusion, we completely removed the mentioning of CFB in this paragraph.

**Line 81: This statement is open to misinterpretation and does not reflect the totality of how these rocks were generated. These rocks are generated by a plume but the melt mechanism is debated. See comments in major points above**
See answers to previous comment and response to major point #3. Alternative models are now included in l. 106 ff.

**Figure 1 - spelling of Kaoko belt is different in the figure.**
Corrected.

**Line 138: this line is unhelpful as it presumes a vector of continuing increasing melt. There is no evidence that underplates form before flows. Indeed, volcanism is contemporaneous with rifting and break up. Underplates may form in response to fractional crystalization at the crust mantle boundary by progressive accumulation of these phases. Delete this line.**
Sentence rephrased.

**Line 141: a typical feature associated with these flows cannot be SDRs as these are seismic features to which the flows themselves belong. Rephrase.**
Sentence rephrased

**Line 146: volcanic not magmatic.**
Corrected.

**Line 148/9: what evidence exists for chemical heterogeneity. No citation is provided and I'm not aware of one in this locale.**
The main factor to distinguish SDR flows from CFB is surely the different prepositional environment. The possible chemical heterogeneity would be reasoned by the different melt source related to a later stage of rifting, compared to the initial CFB signature. We slightly rephrased the sentence to make it clearer, and added a reference, which characterizes SDR's and describes how they may be built by different lava types (l. 174 ff.).

**Line 151: rapidly**
Corrected.

**Line 155: there is evidence of volcanic activity to the north, just much less. The transition isn't as abrupt as noted here. For example, the Namibe basin just north the FFZ has thick SDRs in the south and not much salt. Please examine the existing literature describing the marginal basins to the north of the FFZ.**
The central southern Atlantic section is generally referred to as a magma-poor or non-volcanic passive margin (e.g. Blaich et al., 2011; Contrucci et al., 2004; Mohriak et al., 1990). Of course this does not completely rule out any volcanic activity, which is why we phrased "little to no" magmatic signature. For our models, the strongest reference is the seismic profile corresponding to our marine MT stations presented in Planert et al. (2017). They have interpreted the northern crust as oceanic crust. We follow their interpretation.

**Line 159: citation required for this assertion.**
References added (l. 186 f.).

**Line 162: pronounced**
Corrected.

**Line 163: see paper by Morgan et al 2020 on plume flow in PNAS**
Incorporated the paper (l. 190 ff.)

**Line 395: data do not disclose, rephrase**
Rephrased (l. 459)

**Line 399: comma required**
Comma inserted.

**Line 406: delete further**
Deleted.

**Line 445: Speculation. There is no evidence of particularly wet melts along this margin. Delete.**
We have completely rephrased this hypothesis. It now states that the difference in mantle resistivity could be related to a difference in depletion due to the volatile extraction caused by the hot spot impingement. We removed the references to melt source (l. 509 ff.).

**Line 447/8: it is entirely unclear to the reader how this follows. From my reading of this section, the paper suggests that the speculation of a wetter and drier mantle is associated with more or less plume activity. This is used to suggest the plume is dominant to the north along the WR and that the southern area is 'rift driven breakup'. This is totally unclear as it does not explain the source of magmatisim. Much more discussion is needed and actual evidence from the magmatic system.**

See answer to previous comment.

**References in this response:**

Baba, K., Chen, J., Sommer, M., Utada, H., Geissler, W. H., Jokat, W., and Jegen, M.: Marine magnetotellurics imaged no distinct plume beneath the Tristan da Cunha hotspot in the southern Atlantic Ocean, Tectonophysics, 716, 52–63, 10.1016/j.tecto.2016.09.033, 2017.

Blaich, O. A., Faleide, J. I., and Tsikalas, F.: Crustal breakup and continent-ocean transition at South Atlantic conjugate margins, Journal of Geophysical Research: Solid Earth, 116, 1–38, 10.1029/2010JB007686, 2011.

Contrucci, I., Matias, L., Moulin, M., Géli, L., Klingelhofer, F., Nouzé, H., Aslanian, D., Olivet, J.-L., Réhault, J.-P., and Sibuet, J.-C.: Deep structure of the West African continental margin (Congo, Zaïre, Angola), between 5S and 8S, from reflection/refraction seismics and gravity data, Geophysical Journal International, 158, 529–553, 10.1111/j.1365-246X.2004.02303.x, 2004.

Franz, G., Moorkamp, M., Jegen, M., Berndt, C., and Rabbel, W.: Comparison of Different Coupling Methods for Joint Inversion of Geophysical Data: A Case Study for the Namibian Continental Margin, Journal of Geophysical Research: Solid Earth, 126, 1–28, 10.1029/2021jb022092, 2021.

Gardés, E., Gaillard, F., and Tarits, P.: Toward a unified hydrous olivine electrical conductivity law, Geochemistry, Geophysics, Geosystems, 15, 4984–5000, 10.1002/2014GC005496, 2014.

Gibson, S., Thompson, R., and Day, J.: Timescales and mechanisms of plume lithosphere interactions: 40Ar/39Ar geochronology and geochemistry of alkaline igneous rocks from the Paraná Etendeka large igneous province, Earth and Planetary Science Letters, 251, 1–17, 10.1016/j.epsl.2006.08.004, 2006.

Hirth, G. and Kohlstedt, D. L.: Water in the oceanic upper mantle: implications for rheology, melt extraction and the evolution of the lithosphere, Earth and Planetary Science Letters, 144, 93–108, 10.1016/0012-821X(96)00154-9, 1996.

Jones, A. G., Fullea, J., Evans, R. L., and Muller, M. R.: Water in cratonic lithosphere: Calibrating laboratory-determined models of electrical conductivity of mantle minerals using geophysical and petrological observations, Geochemistry, Geophysics, Geosystems, 13, 10.1029/2012GC004055, 2012.

Mohriak, W., Hobbs, R., and Dewey, J.: Basin-forming processes and the deep structure of the Campos Basin, offshore Brazil, Marine and Petroleum Geology, 7, 94–122, 10.1016/0264-8172(90)90035-F, 1990.

Moulin, M., Aslanian, D., and Unternehr, P.: A new starting point for the South and Equatorial Atlantic Ocean, Earth-Science Reviews, 98, 1–37, 10.1016/j.earscirev.2009.08.001, 2010.

Planert, L., Behrmann, J., Jokat, W., Fromm, T., Ryberg, T., Weber, M., and Haberland, C.: The wide-angle seismic image of a complex rifted margin, offshore North Namibia: Implications for the tectonics of continental breakup, Tectonophysics, 716, 130–148, 10.1016/j.tecto.2016.06.024, 2017.

Strozyk, F., Back, S., and Kukla, P. A.: Comparison of the rift and post-rift architecture of conjugated salt and salt-free basins offshore Brazil and Angola/Namibia, South Atlantic, Tectonophysics, 716, 204–224, 10.1016/j.tecto.2016.12.012, 2017.

Torsvik, T. H., Rousse, S., Labails, C., and Smethurst, M. A.: A new scheme for the opening of the South Atlantic Ocean and the dissection of an Aptian salt basin, Geophysical Journal International, 177, 1315–1333, 10.1111/j.1365-246X.2009.04137.x, 2009.

---

## Author Response (AR2)

GEOMAR | Wischhofstr 1-3 | 24148 Kiel | Germany

**Gesa Franz**
**Doctoral Researcher**

Phone +49 431 600-2556
gfranz@geomar.de

+

**Second Revised Manuscript Submission**

February 1, 2023

Dear Ulrike Werban, dear Editors,

on behalf of all co-authors I re-submit the second revised manuscript entitled *"Formation and geophysical character of transitional crust at the passive continental margin around Walvis Ridge, Namibia"* for publication in Solid Earth.

+

We have received additional comments from one referee, which we have incorporated in the revised manuscript. We hope that the additional modifications to the manuscript clarify any remaining ambiguities. The comments and our answers are appended to this letter.

Again, please don't hesitate to contact us, if you have any further questions.

Kind regards,

Gesa Franz

**GEOMAR**

Helmholtz Centre for
Ocean Research Kiel

Wischhofstr. 1-3
24148 Kiel | Germany

Phone    +49 431 600-0
Fax        +49 431 600-2805
www.geomar.de

Deutsche Bank AG Kiel
BLZ 210 700 24
Kto. 144 8000

SWIFT/BIC
DEUTDEDB210
IBAN DE
69210700240144800000

Tax Number 2029745781

Sales Tax Code
DE281295378

**Foundation under Public Law**
MinDir Volker Rieke, *Chairman of the Board of Governors*
Prof. Dr. Katja Matthes, *Director* | Frank Spiekermann, *Administrative Director*

[Figure]

**Answer to Reviewer #3:**

Thank you, for taking the time to engage in a second review and for your additional comments. We have worked on our manuscript again to include your suggestions and clarify ambiguities.

The Reviewer has mentioned the difficulty to reply to our answers, because their original comments were missing. Therefore, we have now included all of their comments. To facilitate the identification of new answer, we marked all our new responses in green.

Line numbers in this response refer to the tracked revised manuscript, where manuscript alterations are easy to identify.

*Point 1 (Original comment, first review):*

*I may not understand the methods that are used. However a question that might arise for a reader when considering the model was how error was handled. The model that forms the basis of the manuscript examines the electrical resistivity and density at each model cell. But the propagated error calculation that indicates what variability may be caused by uncertainties in the input data to the properties of each model cell is not presented. For a reader that may not be familiar with this type of model, this may be something that is handled within the model generation but for a general readership it might help to describe how such errors are handled so that the reader can have confidence in the conclusions that are being made. For example, in figure 3 the XY plot with individuals symbols would might benefit from x and y error bars in order to assess the distinctness of the clusters. This would therefore permit the reader to assess how different the clusters may be from one another and how robust this differentiation might be.*

*Author response (First review)*

*Handling of errors of the clustering have been discussed in the previous answer to referee #2. As I understand, this issue concerns the errors of the models themselves. The electrical resistivity and density model errors are difficult to derive due to the ambiguity of geophysical inversion. Therefore, it's not a common practice to specify model errors. Usually model anomalies are tested for their necessity, which we performed in the previous publication (Franz et al., 2021). Here, the analysis of models is focused on comparing differences in the models' parameters along the passive margin. The clustering algorithm is used to identify zones of common parameter relationships and distinguishing from zones with different parameter relationships. This zoning is then linked to geological processes to enhance passive margin interpretation.*

*Additional reviewer comment from second review:*

*The authors have addressed the specifics of the clustering mechanism. However, as the authors have correctly interpreted, my query relates to the points on Figure 3. Each point represents an X-Y coordinate that the authors refer to as 'data' in response to the prior comments on error handling of the clustering. Based on the comments above, these points are not data but model outcomes/parameters. The issue I am raising is that of error propagation – one that the authors must address before this work can be published. Each model result has an inherent uncertainty, yet the clustering analysis assumed these to be discrete points with no associated errors (e.g., the points on Figure 3). Accordingly, the statistical treatment of clustering of these points is invalid as it has not propagated the error from the original model. Simply put, how can any value be placed in the clustering when the dimension of potential errors of each point is unknown? Given this is a novel technique, and the probability that this manuscript will be used as a citation for the repetition of this technique elsewhere, it is thus necessary that this issue is fully addressed now lest this issue become a point of contention going forward. I understand that the errors may be complex to handle but perhaps engaging with a statistical specialist may help.*

Our model comprises more than 600,000 model parameters and a single forward calculation can take on the order of an hour. This makes both probabilistic MCMC based approaches as well as traditional SVD based resolution and covariance estimation methods prohibitive. This is reflected in the current literature where the uncertainties of 3D resistivity models is assessed through sensitivity tests as no formal estimate of uncertainty can be given (e.g. Munch & Grayver, 2023; Comeau et al., 2022; Murphy et al., 2023; among many others). We evaluated the reliability of our inversion models by investigating data fit, and various model anomaly sensitivity tests. These tests and synthetic inversions have been described in the previous publication (Franz et al., 2021). We added a description of the two most relevant model uncertainties (l. 244 ff.). These are a conductive model artifact at the northern edge of Walvis Ridge, and the smooth, shallow, conductive anomalies which complicate the differentiation between sediments and upper crustal anomalies. The vertical conductive artifact along the Florianopolis fracture zone is discussed in the manuscript in l. 404 ff. and we refrain from its interpretation as described in l. 630 ff.. The second issue concerning the difficulty to differentiate sediment from crustal anomalies is addressed by integrating seismic constraints to separate the sediment from the crustal domain for clustering and interpretations (l. 253 ff.).

While it is important to have some understanding of the variability of the model parameters as discussed in our manuscript, the purpose of the cluster analysis is not to derive statistical properties of the clusters. Instead we use the analysis as approximation of geological units with different properties. Any reader familiar with the geological interpretation of geophysical results will understand that the plotted domains will have some uncertainty even if a formal estimate of this uncertainty cannot be given. We also added references to articles where the authors have performed similar clustering analyses (called geology differentiation) based on models resulting from geophysical inversion (l. 236 f.).

***Point 3: Original Comment:***
*A major issue in this manuscript is the discussion as it relates to the mantle. The manuscript asserts that the difference in resistivity of the mantle relates primarily to differences in depletion associated with a mantle plume. Specifically, the increased magma generation associated with the plume reduced the iron and hydrogen content of the residual mantle, thus increasing the resistivity. This hypothesis relies on the assumption that i) magma generation south of the Walvis Ridge is from melting of an upper mantle without the significant influence of a plume. This concept is alluded to earlier in the manuscript on line 79 where it is suggested that the continental flood basalts in this region are 'mainly of upper mantle composition instead of a deep plume' and also later on line 447/8 (see comments in the line by line). ii) Magma generation at the Walvis Ridge area is the result of plume melt. These assumptions may be problematic:*
*A) The origin of continental flood basalts in this region is not universally considered to be in the shallow upper mantle (i.e., lithospheric mantle) as suggested in the manuscript. While some authors argue for this source as correctly pointed out by the citation used, others present counter arguments. Please read and incorporate the following citations:*
*Thompson, R.N., Gibson, S.A., Dickin, A.P., and Smith, P.M., 2001, Early Cretaceous basalt and picrite dykes of the southern Etendeka region, NW Namibia: windows into the role of the Tristan mantle plume in Paraná–Etendeka magmatism: Journal of Petrology, v. 42, p. 2049–2081.*
*Ewart, A., Marsh, J.S., Milner, S.C., Duncan, A.R., Kamber, B.S., and Armstrong, R.A., 2004, Petrology and geochemistry of Early Cretaceous bimodal continental flood volcanism of the NW Etendeka, Namibia. Part 1: Introduction, mafic lavas and re-evaluation of mantle source components: Journal of Petrology, v. 45, p. 59–105.*
*Gibson, S.A., Thompson, R.N., and Day, J.A., 2006, Timescales and mechanisms of plume–lithosphere*

*interactions: 40Ar/39Ar geochronology and geochemistry of alkaline igneous rocks from the Paraná–Etendeka large igneous province: Earth and Planetary Science Letters, v. 251, p. 1–17.*
*B) Plume sources are considered to have more water and iron than the depleted upper mantle – please research works by Dixon and also Herzberg. While melting of a plume source may lead to depletion, it would require all the material to have been melted. There are further questions on this model as noted below.*
*C) The depth over which the model is sensitive is ~300km, and at least 100km is being interpreted in the manuscript - as presented per the manuscript text and figures. This extends below the thinned lithospheric mantle along this continental margin and is within the convecting upper mantle. This would suggest that melt depleted mantle material has remained within the convecting upper mantle over an extended interval. The manuscript does not present a mantle flow field argument supporting that this is possible. Moreover, the upper 300km of mantle in the region has seen material from the African LLSVP intrude into it (see recent paper by O'Connor Nature Communications in 2020). Melting of such material may not occur until about 120km depth if the mantle potential temperature is 1530C. This would result in a complex mantle with residual and enriched materials. How might hybrid compositions of pyroxenites impact the interpretations of the model?*

*On the basis of these points, the hypothesis posed in the manuscript is interesting but requires further support and clarification.*

***Response from authors Part 1***
*In our manuscript we state the hypothesis, that the crustal structure south of Walvis Ridge and along the ridge differ as a result of the direct plume impact at Walvis Ridge latitudes. We note, that the involvement of the Tristan plume is a topic to debate (l. 99 f.). This comment helped us to realize inconsistencies in our hypothesis. What we actually wanted to state, is a difference in the crust and upper mantle related to the degree of mantle and melt depletion. And to link the higher depletion below Walvis Ridge to the impingement of the Tristan hot spot and the corresponding extraction of volatiles. The residual upper mantle would then be more depleted compared to the hypothesized "rift related" crust south of Walvis Ridge. We have rephrased the parts of the discussion and conclusion to clearly describe this hypothesis (l. 509ff. and 665 ff.).*

***Reviewer comment on revised manuscript - Part 1***
*The response has helped clarify the manuscript but the adjusted line in the conclusion retains the binary view of plume/rift. For example line 666 (the line numbers quoted in the response are not correct) states "Our hypothesis is, that these variations in the mantle composition may result from different degrees of mantle depletion, linked to the differentiation between a rift-related southern complex, and a plume-driven Walvis Ridge regime." This perpetuates the idea that the magmatism in the south has no plume influence – in contrast with the data in the papers I cited previously. If the authors wish to state that the presence of the Tristan tail in particular enhanced melting in the region in question, that would be fine but the current binary of plume-rift is inaccurate.*

We removed the binary interpretation of a rift- and plume related differentiation. Instead we ascribe the difference between the southern and along Walvis Ridge domain to the increasing volcanic activity at Walvis Ridge, related to the impingement of the Tristan plume (l. 675 ff.).

***Response from authors - Part 2***
*We are not specialists in isotope geochemistry, but have evaluated the proposed papers. We believe that their conclusions do not contradict our statements. We state that the earliest phase of continent break-up is associated with rifting and that that early magmatics are mainly of upper mantle composition (in*

*l.100 f.). Gibson et al. (2006) also link this earliest stage of the CFB emplacement (~145 Ma) to melts at the mechanical boundary layer (MBL) at ~150 km depth and not a deep plume source. We do not rule out involvement of plume material in the Etendeka CFB in the subsequent stages. In fact we point out the interaction of the Tristan plume and the lithosphere and heterogeneous composition (l. 131 ff.) of intrusive magmatics, which we link to the ascend of magma which forms dykes and eventually the CFB (l. 136, 142).*

**Reviewer comment on revised manuscript - Part 2**
*It might be helpful to seek the input from the many isotope geochemists at GEOMAR – they also work in the same region. The conclusions in Gibson do indeed create difficulties if it assumed that magmatic activity south of the Walvis ridge is restricted to a rift and decompression melt of the ambient upper mantle. That manuscript presents a model whereby the alkaline activity is generated by conductive melting of a lithospheric mantle with a proposed 250 degree heat differential from ambient (a plume). Accordingly, the melt is initially caused by the thermal influence of the plume (and why the initial melt is at the MBL). As the lithosphere thins large scale melting of this mantle with elevated temperatures occurs. The authors state in their response that 'do not rule out involvement of plume material in the Etendeka CFB in the subsequent stages' – however this is precisely what is implied by the conclusion sentence in the revised version noted above. This could all be clarified if the authors used less binary terms and making it clearer that there was a more significant plume influence along the Walvis Ridge area. However, more on this issue below.*

As noted above, we removed the binary interpretation of a rift- and plume related differentiation.

**Response from authors Part 3**
*Concerning the comment about the model depth and depth of depleted mantle: Thank you for describing this problem of a mismatch of mantle convection and our statements about a different mantle structure south of-, and along Walvis Ridge. We understand that there needs to be clarification, because we haven't clearly stated that interpretations should be confined to the upper/lithospheric mantle only. We added appropriate statements: We point out, that the resolution capabilities of the electrical resistivity model decrease with depth, and the statements therefore become more vague with depth (l. 635 f.). Additionally, we clearly phrased that interpretations of the mantle domain should not extent below the LAB in l. 389 ff. In our discussion of the mantle clusters, we also added the explicit statements, that our interpretations concern the shallow, lithospheric mantle (l. 513 f. and 562 f.).*

**Reviewer comment on revised manuscript - Part 3**

*It is good to see this clarified but it isn't clear everywhere. For example, the line in the conclusion states only 'mantle' and not lithospheric mantle. Everywhere 'mantle' is mentioned it must be changed to lithospheric mantle throughout the document. Otherwise, confusion will continue for those only reading the paper quickly. However, this now brings up an important question – what is the nature of the depleted lithospheric mantle that the authors are referring to? Ostensibly, the authors suggest that the depleted lithospheric mantle in the Walvis Ridge domain relates to plume related-melting. However, this lithospheric mantle in this region should be residual continental lithospheric mantle and thus has been depleted by melting associated with a change in the geotherm (see Gibson paper above for the mechanism). If this is the case, then depletion of the lithospheric mantle by melt creation from within it is a widespread process and not limited to the Walvis Ridge area. The manuscript is very unclear on this point and needs to consider what exactly has been depleted and how.*

*On line 561 of the new manuscript is the following:*

*"We attribute these high mantle values to the remnant signature of the upwelling plume, where volatile elements are extracted from melts and rise to the surface to form flood basalts, volcanic flows, and the new oceanic crust (Mutter et al., 1988). The depleted material left in the shallow, lithospheric mantle is highly resistive due to the lack of fluid phases and elements like iron and hydrogen (Baba, 2005; Evans et al., 2005; Matsuno et al., 2010; Selway, 2014)."*

*This line is relevant to the points above as it explains what the authors' model is. Firstly, there are serious issues with shifting topics in this sentence that make it ambiguous. As written, this sentence implies that volatile element are extract from melts. I suspect the authors mean 'by melts'. Moreover, the 'and rise' should be 'that rise'. From this model, it would seem that the lithospheric mantle in this region is residual from the plume and not residual continental lithospheric mantle. Please clarify.*

We added the term lithospheric to all occurrences of interpretations concerning the mantle and altered the sentence in l. 568 ff. according to the suggestions. Additionally, we expanded the description of the above stated sentence (l. 568 ff.). In our model, the emerging of the Tristan plume leads to large amounts of mantle melts, intra-crustal magmatism, crustal thickening, uplift, and surface volcanism. All of these features result in a significantly high crustal electrical resistivity. The depletion of the underlying lithospheric mantle is a result of this increased magmatic/volcanic activity and also results in high upper lithospheric mantle resistivities. Our analysis of large scale electrical resistivity and density variations is not eligible for further interpretation regarding the mantle composition or origin.

**Point 4: Original Comment**
*An additional area of concern relates to the conductivity measurements in the upper crust north of the Walvis ridge. This region is known to have significant salt deposits. There is no discussion of the impact of even small salt horizons in this region. There is an allusion to this with respect to highly conductive layers, for example associated with mineralization of lavas. However, it wasn't apparent that any discussion has occurred in relation to these already mapped salt horizons. The authors must address this directly in their models as workers in this region will be familiar with these deposits and it would raise questions that would detract from this important work.*

**Response from authors**
*Salt deposits north of Walvis Ridge have been mapped offshore Angola in the Kwanza basin north of ~15°S (e.g. Blaich et al., 2011; Moulin et al., 2010; Strozyk et al., 2017, Torsvik et al., 2009). The salt directly adjacent to the FFZ may have been sheared off to the South American margin during the Albian ridge jump. The latitudes north of 15°S are not included in our model area. Therefore, we do not discuss any inclusion of salt horizons in our model region.*

**Revised Manuscript Comment**
*There is salt in the basin directly north of the Walvis Ridge (and this basin is very much not north of 15S). The Namibe basin has been mapped as having 0-70m of "Evaporites – gypsum and anhydrite. Halite in subsurface" by Jerram et al., 2019 (doi:10.1016/j.tecto.2018.07.027)*

*The authors will need to address the potential for salt in the crustal rocks and the implications on the observations and potential vertical smearing of such highly conductive units. While some authors have interpreted there to be no salt based on the seismic lines, the physically mapped rocks show these interpretations to likely be erroneous. The magnitude of the salt is much reduced in comparison to the north, leading to the potential of it not being detected with seismic methods. However, given the*

*sensitivity of MT to such deposits, it is important to assess the potential for this material in the sedimentary layers of the model.*

The named reference (Jerram et al., 2019) describes the 0-70 m "Evaporites – gypsum and anhydrite. Halite in subsurface" sequence in the Kwanza and Namibe basins. In their figure 1, the location of the three study regions: Kwanza, Benguela and Namibe basins are mapped. The southern most basin is mapped slightly north of Port of Namibe. This port is located in the city Moçâmedes, which has the coordinates of 15°11'S, 12°07'E. Our northernmost MT stations is located at 18°08'S, 9°56'E.

We do not see any reference for salt layers in our electrical resistivity models, which should be high electrical resistivity, or in the conjoined seismic velocity models by Planert et al. (2016). A density model which we have previously used as reference (Maystrenko et al., 2013) does not mention any salt in the area, either. The seismic profiles in Strozyk et al. (2017) image small amounts of evaporites in the northern Namibe basin (latitude ~15°S), and explicitly none in the southern Namibe basin (latitude ~17.5°S) (their figures 1 and 7). Therefore, we still do not see any need to address the salt basins along the Angolan coast.

**Line by line**

*New MS Line 174: "The different depositional environment and possibly variable chemical composition due to a different melt source, distinguishes them from the initial continental flood basalts (McDermott et al., 2018)"*

*This line was changed in relation to my comment:*
*"what evidence exists for chemical heterogeneity. No citation is provided and I'm not aware of one in this locale."*

*The author response was "The main factor to distinguish SDR flows from CFB is surely the different prepositional environment. The possible chemical heterogeneity would be reasoned by the different melt source related to a later stage of rifting, compared to the initial CFB signature. We slightly rephrased the sentence to make it clearer, and added a reference, which characterizes SDR's and describes how they may be built by different lava types (l. 174 ff.).*

*McDermott presents no chemical data to distinguish the composition of the SDRs from the CFBs. Indeed McDermott uses inference to suggest the continued influence of a plume in the SDRs in the South Atlantic. Reference to chemical or compositional differences must be deleted unless the authors can provide an appropriate citation supporting this assertion.*

Reference to chemical composition removed (l. 175).

*Line 181 – "While thickened crust and the features described above (magmatic underplating, periodic magmatic flows, and magmatic dykes) characterize the COT zone south of Walvis Ridge, the crust north of the FFZ is distinctly thinner, with little to no magmatic signature (Aslanian et al., 2009; Blaich et al., 2011; Planert et al., 2017)."*

*Original comment "there is evidence of volcanic activity to the north, just much less. The transition isn't as abrupt as noted here. For example, the Namibe basin just north the FFZ has thick SDRs in the south and not much salt. Please examine the existing literature describing the marginal basins to the*

*north of the FFZ.”*

*Author Response: “The central southern Atlantic section is generally referred to as a magma-poor or non-volcanic passive margin (e.g. Blaich et al., 2011; Contrucci et al., 2004; Mohriak et al., 1990). Of course this does not completely rule out any volcanic activity, which is why we phrased “little to no” magmatic signature. For our models, the strongest reference is the seismic profile corresponding to our marine MT stations presented in Planert et al. (2017). They have interpreted the northern crust as oceanic crust. We follow their interpretation.”*

*Revised Response: This interpretation conflicts with the cited paper by McDermott et al., 2018, who suggest that the flows of the Namibe basin are Type I SDRs. Also see Figure 7 of Strozyk et al., 2017 (10.1016/j.tecto.2016.12.012) who show SDRs in the southern Namibe basin. This is a far more complicated situation than the authors are portraying*

We altered the description of the geological setting north of Walvis Ridge to emphasize, that the area is less affected by magmatic/plume overprint (l. 181 f. and l. 200). That does not rule out magmatic signatures in general and we have not excluded such signatures. The main point here is a major difference of the crust south of Walvis Ridge and north of it. From all mentioned references, it is clearly evident that the crustal structure changes drastically over the Florianopolis fracture zone.

References in this response:

Comeau, M. J., Becken, M., Grayver, A. V., Käufl, J. S., & Kuvshinov, A. V.: The geophysical signature of a continental intraplate volcanic system: From surface to mantle source. Earth and Planetary Science Letters, 578, 117307. https://doi.org/10.1016/j.epsl.2021.117307, 2022.

Franz, G., Moorkamp, M., Jegen, M., Berndt, C., and Rabbel, W.: Comparison of Different Coupling Methods for Joint Inversion of Geophysical Data: A Case Study for the Namibian Continental Margin, Journal of Geophysical Research: Solid Earth, 126, 1–28, 10.1029/2021jb022092, 2021.

Jerram, D. A., Sharp, I. R., Torsvik, T. H., Poulsen, R., Watton, T., Freitag, U., Halton, A., Sherlock, S. C., Malley, J. A. S., Finley, A., Roberge, J., Swart, R., Puigdefabregas, C., Ferreira, C. H., & Machado, V.: Volcanic constraints on the unzipping of Africa from South America: Insights from new geochronological controls along the Angola margin. Tectonophysics, 760, 252–266. https://doi.org/10.1016/j.tecto.2018.07.027, 2019.

Maystrenko, Y. P., Scheck-Wenderoth, M., Hartwig, A., Anka, Z., Watts, A. B., Hirsch, K. K., and Fishwick, S.: Structural features of the Southwest African continental margin according to results of lithosphere-scale 3D gravity and thermal modelling, Tectonophysics, 604, 104–121, 10.1016/j.tecto.2013.04.014, 2013.

Munch, F. D., & Grayver, A.: Multi-scale imaging of 3-D electrical conductivity structure under the contiguous US constrains lateral variations in the upper mantle water content. Earth and Planetary Science Letters, 602, 117939. https://doi.org/10.1016/j.epsl.2022.117939, 2023.

Murphy, B. S., Bedrosian, P. A., & Kelbert, A.: Geoelectric constraints on the Precambrian assembly and architecture of southern Laurentia. In Laurentia: Turning Points in the Evolution of a Continent (pp. 203–220). Geological Society of America. https://doi.org/10.1130/2022.1220(13), 2023.

Planert, L., Behrmann, J., Jokat, W., Fromm, T., Ryberg, T., Weber, M., and Haberland, C.: The wide-angle seismic image of a complex rifted margin, offshore North Namibia: Implications for the tectonics of continental breakup, Tectonophysics, 716, 130–148, 10.1016/j.tecto.2016.06.024, 2017.

Strozyk, F., Back, S., & Kukla, P. A.: Comparison of the rift and post-rift architecture of conjugated salt and salt-free basins offshore Brazil and Angola/Namibia, South Atlantic. Tectonophysics, 716, 204–224. https://doi.org/10.1016/j.tecto.2016.12.012, 2017.